# Regular physical activity affects brain activities in old individuals: an observational study

**Keisuke Fukasawa[1]◉, Hideyuki Hoshi[2,3]◉, Yoko Hirata[4], Momoko Kobayashi[3], Keita Shibamiya[3], Sayuri Ichikawa[1], Yoshihito Shigihara** [2,3]*

1 Clinical Laboratory, Kumagaya General Hospital, Kumagaya, Saitama, Japan, 2 Precision Medicine Centre, Hokuto Hospital, Obihiro-shi, Hokkaido, Japan, 3 Precision Medicine Centre, Kumagaya General Hospital, Kumagaya, Saitama, Japan, 4 Department of Neurosurgery, Kumagaya General Hospital, Kumagaya, Saitama, Japan

◉ These authors contributed equally to this work.
* y-shigihara@hokuto7.or.jp

## Abstract

### Background

A healthy lifestyle, including regular physical activity, prevents cognitive decline and dementia. Evaluating the influence of regular physical activity on the brain is essential for properly assessing patients' conditions and designing effective therapeutic strategies. We aimed to investigate whether and how electrophysiological brain activity reflects the influence of regular physical activity.

### Methods and Findings

Clinical records from 327 patients who visited our outpatient department for dementia were analysed retrospectively. Patients were classified into two groups: 'Active' for those who engaged in regular physical activity and 'Nonactive' for patients who did not. Electrophysiological brain activity was recorded using magnetoencephalography and quantitatively evaluated using three spectral parameters: median frequency, individual alpha frequency, and Shannon's spectral entropy. Cognitive state was assessed using three neuropsychological assessments: the Japanese version of Mini-Mental State Examination (MMSE-J), Frontal Assessment Battery (FAB-J), and Alzheimer's Disease Assessment Scale-Cognitive section (ADAS-J cog). The effects of group ('Active' or 'Nonactive') on the spectral parameters were examined using an analysis of covariance with one of the neuropsychological assessments as a covariate. The size of contribution was quantified in the unit of neuropsychological assessments using a regression model. A main effect of group was observed for all three spectral parameters. The size of contribution was equivalent to approximate changes of 3–11 points in MMSE-J, 3–7 points in FAB-J, and 10–14 points in ADAS-J cog scores. The main limitations of our study are: (1) this study was conducted in a single

**Data availability statement:** All data are available in a matlab file from the following repository: Shigihara, Yoshihito; Hoshi, Hideyuki; Fukasawa, Keisuke (2024), "Regular physical activity affects brain activities", Mendeley Data, doi: https://doi.org/10.17632/xgzgpk9475.1 (https://doi.org/10.17632/xgzgpk9475.1)

**Funding:** This study was partially supported by RICOH Co., Ltd. The funders had no role in study design, data collection and analysis, decision to publish, or preparation of the manuscript.

**Competing interests:** YS is leading joint research projects with RICOH Co., Ltd. and Itoen Co., Ltd. HH was employed by RICOH Co., Ltd. KF, YH, MK, KS, and SI declare no conflict of interest.

site; (2) possibility of reverse causality; and (3) some potential confounding factors, such as genetic factors, were not considered.

## Conclusions

Electrophysiological brain activity reflects the influence of regular physical activity as well as current cognitive states. Such insights are valuable for physicians to design effective therapeutic strategies and provide clinical advice to patients with cognitive impairment and dementia.

## Introduction

Dementia is a functionally defined 'state' where various brain diseases cause cognitive impairments that interfere with daily activities, rather than 'pathological conditions' such as neural death because of amyloid deposition or hippocampal atrophy. Cognition comprises a broad range of brain functions, including perceiving, thinking, knowing, reasoning, remembering, analysing, planning, paying attention, generating, synthesising ideas, creating, judging, being aware, and having insight [1]. This process is referred to as the cognitive state [2] and can be impaired by various factors, such as diseases [3–5]. However, the severity of cognitive impairments does not always reflect the severity of causative pathologies [6] because lifestyle-associated factors, such as less education, head injury, physical inactivity, smoking, excessive alcohol consumption, hypertension, obesity, diabetes, hearing loss, depression, infrequent social contact, and air pollution, interfere with their relationships [7]. These factors and their modifications influence pathological conditions (e.g., neurodegeneration, inflammation, and vascular damages) and some drive physiological compensation mechanisms (e.g., axonogenesis, synaptogenesis, neural plasticity, flexibility, and efficiency) [7–11], which consequently contribute to preventing future cognitive decline as well as maintaining the current cognitive state. The state itself is a treatment target and patients' interest; thus, its precise and detailed assessment is essential. In clinical practice, the cognitive state is assessed through careful interviews with physicians, supported by a battery of medical examinations and assessments, reflecting its multi-dimensionality. Neuropsychological assessments, such as the Mini-Mental State Examination (MMSE) [12], Frontal Assessment Battery (FAB) [13], and Alzheimer's Disease Assessment Scale-Cognitive section (ADAS cog) [14], are the most essential and frequently used examinations to evaluate dementia-associated cognitive states. Although these assessments have been well established [15], they are not always reliable because various factors, such as practice effects [16,17], ceiling and floor effects [18], and physical disabilities [19], influence their scores.

To improve the accuracy and validity of clinical diagnoses, other assessments of the cognitive state have been proposed, which can be used complementarily to the neuropsychological assessments. Electrophysiological assessments, including magnetoencephalography (MEG) and electroencephalography, are candidate modalities. Both are non-invasive clinical tools to evaluate neurological states in terms of

electrophysiological brain activity, whose changes reflect brain functions that are indicative of dementia-associated cognitive states [20–23]. The characteristics of electrophysiological activities can be summarised in various ways [20–23]. In this study, we focused on spectral parameters, such as the median frequency (MF), individual alpha frequency (IAF), and Shannon's spectral entropy (SSE), which are sensitive to cognitive decline. They capture spectral changes in electrophysiological brain activities, which are known indicators of cognitive impairments; for example, (1) enhanced low-frequency oscillatory activity accompanied by attenuated high-frequency oscillatory activity, (2) slowing down of the alpha peak frequency, (3) less prominent alpha oscillations, and (4) loss of diversity of neural oscillatory components [24–27]. The MF, IAF, and SSE can represent these changes concisely [28–30]. Lower values of these parameters are associated with lower cognitive states [20,23,25,29–38] and a higher plausibility of dementia, consistent with clinical impressions [34]. Since 2019, we have routinely used these three MEG spectral parameters in our outpatient dementia department. These values are used during medical interviews and consultations with neuropsychological assessment scores to provide better treatments and care [25,34]. We have previously shown that they are well-correlated with neuropsychological assessment scores, namely current cognitive states [25]. Furthermore, they also carry information regarding pathological states, which are usually measured using positron emission tomography, single photon emission tomography, and ultrasonography, and are considered drivers of cognitive impairments [35,39–42]. These findings suggest that MEG spectral parameters capture broad information about neurological states, including physiological, neuropsychological (i.e., current cognitive states), and pathological conditions (Fig 1). Lifestyle-associated factors modify the potential risk of cognitive decline and current cognitive state by changing the neurological state in two ways: via anti-pathological and physiological compensation pathways [7–9]. Therefore, the MEG spectral parameters should also capture the neurological changes which do not influence the current cognitive state (i.e., neuropsychological assessment scores) but withstand potential neurologically-accumulated risks of cognitive decline (Fig 1). We hypothesise that the MEG spectral parameters would be indicative of neurological changes triggered by lifestyle-associated factors; thus, they should differ between patients with different lifestyles and correlate with neuropsychological assessment scores (i.e., current cognitive states). If the hypothesise is correct, objective evaluation of the lifestyle-associated factors using MEG assist physicians in designing therapeutic strategies and promoting lifestyle changes to prevent cognitive decline. To examine the hypotheses, we assessed the effects of lifestyle-associated factors on MEG spectral parameters while controlling for the effects of neuropsychological assessment scores.

In the present study, we focused on regular physical activity as a representative index of lifestyle-associated factors, without excluding influences from other lifestyle-related factors. We chose regular physical activity because of its clinical importance. As one of the few prospectively modifiable factors (e.g., by encouraging patients during medical consultation), it is worth addressing. In contrast, other factors, such as education and air pollution, are not easily modifiable for current patients; thus, the benefits of discussing those are limited for the clinical practice. We retrospectively analysed clinical data from 327 patients and compared the MEG spectral parameters between patients with and without regular physical activity, while controlling for neuropsychological scores to extract additional information associated with regular physical activity.

## Materials and methods

### Patients and ethics

This retrospective observational study used a large clinical dataset of 327 patients (188 women and 139 men; mean age±standard deviation, 77.9±7.1 years [range 49–93 years]) who visited our outpatient department for dementia. The inclusion criteria were patients visiting the department for the first time between 5 June 2019 and 11 April 2024. The exclusion criteria were as follows: (1) lack of documentation of regular physical activity, (2) refusal to consent for data reuse, and (3) not undergoing MEG. According to the clinical records, 32 were diagnosed with healthy ageing, 60 with mild cognitive impairment, and 220 with dementia. Thirteen were diagnosed with other medical conditions such as depression or disuse syndrome. Two patients did not present on the day of diagnosis. We included all patients who met the

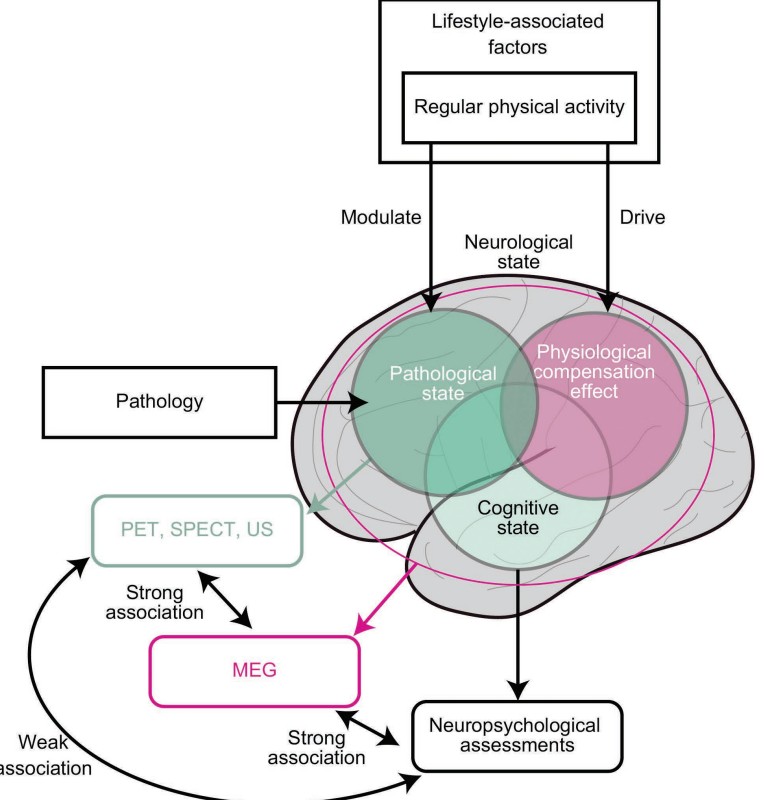

**Fig 1. A Model of the Effects of Lifestyle-Associated Factors on Cognitive States.** Lifestyle-associated factors, such as regular physical activity, modulate the pathological state and drive physiological compensation mechanisms, which jointly influence the current cognitive state. While neuropsychological assessments capture cognitive states and pathological measurements (e.g., positron emission tomography [PET], single photon emission tomography [SPECT], and ultrasonography [US]) capture the pathological state, magnetoencephalography (MEG) (i.e., electrophysiological measurements) captures broad neurological conditions. Therefore, we expected that it would capture the comprehensive effects of lifestyle-associated factors on neurological conditions.

aforementioned criteria, regardless of their diagnosis, because we would like to apply our findings to all patients who seek lifestyle intervention. This study was approved by the Ethics Committee of Kumagaya General Hospital (#34) and adhered to relevant Japanese guidelines and regulations. All data used for the present study were anonymised at the first stage. Only two of the authors (KF and YS) had the ability to identify individual patients using comparative tables, if necessary. Data were accessed for research purposes between 31 May and 5 July 2024. All participants provided written informed consent to participate in this study if they were cognitively healthy. Otherwise, their legal guardians (i.e., family members) provided informed consent on their behalf. Because all patients visited the outpatient department for dementia and any clinical problems were suspected to be cognition-related, we obtained written informed consent from both the patients and their legal guardians, except for special circumstances. The procedure was performed in accordance with the Ethical Guidelines for Medical and Health Research Involving Human Subjects, published by the Japanese Ministry of Education, Culture, Sports, Science, and Technology.

## Classification of active and nonactive groups

Documentation of regular physical activity was obtained from clinical records written by the attending physician (Author YH) during medical interviews. Patients were classified as 'Active' if their records indicated any regular exercise routine,

such as daily walking or participation in a gymnastics club, whose amount, duration, and frequency were at a reasonable level considering each patient's background judged by the physician. Those without such a routine were classified as 'Nonactive'. Patients whose records did not describe regular exercise routines were excluded. We intentionally did not set thresholds for the classification of 'Active' and 'Nonactive' across participants, such as types of activities (e.g., walking, swimming, playing golf, and trekking), duration (e.g., two hours per day), and frequency (e.g., every two days), because the definition of 'Active' varies depending on patients' backgrounds. As described in a guideline by the World Health Organisation for preventing cognitive impairment and dementia, even small amounts of physical activity can be considered as an exercise for some [43]. Smaller than recommended amounts of physical activity can still benefit cognition [44]. Old individuals often suffer from physical disabilities such as dizziness or pain in their hip or knee. Even 2,000–3,000 steps of walking every two days represents a significant effort for them. While we may deem this a small amount of exercise for us, it is significant for them and improves their physical/mental condition and cognition. Due to the diversity in the clinical population, it was neither practical nor fair to apply uniform thresholds for the classification of 'Active' and 'Nonactive' groups. Instead, despite being subjective, the patients were classified by a physician, considering their daily activity status and their clinical background.

## Neuropsychological assessments

Regarding the cognitive state, we employed three neuropsychological assessment scores that are routinely used in our department: the total scores of the Japanese versions of the MMSE (MMSE-J) [45,46], Frontal Assessment Battery (FAB-J) [13,47], and Alzheimer's Disease Assessment Scale-Cognitive section (ADAS-J cog) [48]. MMSE-J and FAB-J scores range from 0–30 and 0–18, respectively, with lower scores indicating more severe impairment. ADAS-J cog scores range from 0–70, with higher scores indicating more severe impairment. MMSE-J scores were missing for one patient, FAB-J scores for five patients, and ADAS-J cog scores for 65 patients because of clinical limitations.

## MEG data acquisition

To quantify electrophysiological brain activity, three spectral parameters—MF, IAF, and SSE—were retrieved from clinical records [30,36]. They had been computed from five minutes of resting-state brain activity (i.e., spontaneous neural oscillatory activity). The activities were recorded using a 160-channel whole-head magnetoencephalography system (RICOH160−1; RICOH, Tokyo, Japan) in a magnetically shielded room as part of clinical practice. Patients were asked to remain calm in the supine position with their eyes closed during the scan. The sampling frequency was 2000 Hz with 500 Hz low-pass filtering during the recording.

## MEG analysis

The three spectral parameters—MF, IAF, and SSE— had been computed using the same sensor-level analysis protocol used in our previous studies and those of other groups [25,29–31,34,36–38,40,49–51] during daily clinical practice at Kumagaya General Hospital. MEG data are sometimes contaminated by artefacts such as signal fluctuation caused by dental works, which can deteriorate the quality of analysis. Before starting the analyses, artefacts were manually removed using principal component analysis, if necessary, by experienced physicians or clinical laboratory technicians (Authors FK, SI, and YS). The artifact removal was conducted using RICOH MEG Analysis software (RICOH, Tokyo, Japan), an analysis software provided by the MEG manufacturer. The technique was applied to 125 of 207 MEG datasets as part of clinical practice. The number of removed components were adjusted for each dataset and limited to as few as necessary. Subsequent MEG analyses were performed offline using MATLAB (MathWorks, Natick, MA, USA). The time-series signals were band-pass (1–70 Hz) and band-stop filtered at 50 Hz to remove the power line noise. The power spectral density (PSD) was computed using the Blackman-Tukey approach [52] with non-overlapping 5-s segments of the filtered signals. In the Blackman-Tukey method, PSD is computed as a discrete Fourier transform of the autocorrelation function of the

filtered time-series signals, which has better precision than other approaches [53] and is commonly used for computing MEG spectral parameters [25,30,31,34–37,50]. The original PSD was divided by the total power in the frequency range of interest (1–70 Hz) and the normalised PSD (PSDn) was obtained.

Three spectral parameters (i.e., MF, IAF, and SSE) were calculated to summarise different characteristics of the PSDn [20,23,25,29–32,34–38,40,49,50]. The MF quantifies the frequency at which the spectral power is balanced between low and high frequencies. It divides the PSDn into two equal halves between 1 and 70 Hz. The IAF represents the dominant frequency corresponding to the peak of the PSDn in the alpha band. It is defined similarly to MF, but with an adjusted frequency range between 4 and 15 Hz (i.e., extended alpha band) instead of 1–70 Hz, to obtain a robust estimator of the dominant alpha oscillations. SSE is computed as the normalised Shannon's entropy to the PSDn, which can be calculated as a probability density function:

$$\text{SSE} = -\frac{1}{\log(N)} \cdot \sum_{f=1Hz}^{70Hz} PSDn(f) \log[PSDn(f)]$$

(1)

where $N$ is the number of frequency bins of the PSDn. SSE represents an irregularity measure closely related to the concept of order in information theory, which quantifies the homogeneity in the distribution of the oscillatory components of the PSDn. Sensor- and epoch-wise MF, IAF, and SSE were computed. Then, they were averaged across all sensors and epochs. These parameters have been used in clinical practice at some hospitals [25,34], with lower values indicating a lower cognitive state, and are associated with neuropsychological assessment scores [20,23,25,29–38].

Changes in the three spectral parameters (i.e., MF, IAF, and SSE) reflect changes in patients' cognitive states [34] and neurophysiological conditions in the brain, such as changes in excitation/inhibition ratios and neuromodulator release. Regarding cognitive states, IAF and SSE are associated with MMSE and FAB scores, respectively, while MF is well associated with both MMSE and FAB scores [25]. This discrepancy suggests that three MEG spectral parameters are associated with distinct cognitive functions. Regarding neurophysiological conditions, low-frequency oscillatory activity (e.g., delta and theta) is associated with ascending cholinergic input to cortices from subcortical regions, such as the nucleus basalis of Meynert [54,55]. Amyloid deposit in the cortex also enhances the low-frequency oscillatory activity [39]. Stroke also enhances low frequency oscillatory activity [56]. Clinically, pathological structural changes often influence on its amplitude. Alpha oscillatory activity, which is considered an intermediate frequency, is produced by the thalamocortical network [57] and plays a key role in cognition [58]. Although its frequency is distributed from 8 to 12 Hz, it slows down in patients with mild cognitive impairment [59] and Alzheimer's disease [60]. Its power attenuates in patients with Alzheimer's disease [61] and dementia due to Lewy body [62]. High-frequency oscillatory activity (e.g., beta and gamma) is associated with gamma-aminobutyric acid level [63] which is a inhibitory neurotransmitter [64]. It is related to synaptic activities [65], network activities [66], and neuroplasticity [67]. In the predictive cording theory [68], low-frequency oscillatory activity is associated with top down processing in the brain, while high-frequency oscillatory activity is associated with top down processing bottom-up processing [69]. MF represents the power balance between wide frequency range from delta to gamma, while IAF just focus on frequency balance around alpha which is sensitive to frequency change in peak alpha frequency. SSE covers the same frequency range but evaluates different aspects of PSD: entropy rather than a simple power balance between high- and low-frequency oscillatory activities.

## Statistical analysis

Statistical analyses were performed using MATLAB (MathWorks, Natick, MA, USA). Given the large number of samples (N = 327), we used parametric statistical tests for all analyses. First, to examine the effect of regular physical activity on cognitive states and electrophysiological activities, we examined group-level differences in each neuropsychological score, MEG spectral parameter, and age between the two groups ('Active' and 'Nonactive') using two-sample t-tests.

Next, to explore the relationships within/between cognitive states and electrophysiological activities, Pearson's correlation coefficients were calculated to investigate the intra- and intercorrelations within/between the age, neuropsychological scores, and MEG spectral parameters for each group. The correlation coefficients were tested with the null-hypothesis of no-correlation against the alternative hypothesis of a nonzero correlation. For the $t$-tests and correlation analyses, $p$-values were adjusted for false discovery rate (FDR) using the Benjamini–Hochberg method [70], with significance set at less than 0.05. Next, we assessed the effect of regular physical activity on MEG spectral parameters while accounting for cognitive state using analysis of covariance (ANCOVA). Each MEG spectral parameter (MF, IAF, and SSE) was considered as a dependent variable, while regular physical activities ('Active' and 'Nonactive') was used as the independent variable and age or one of the neuropsychological assessment scores (MMSE-J, FAB-J, or ADAS-J cog) was used as a covariate.

Finally, the impact of regular physical activity on MEG spectral parameters was quantified in the unit of neuropsychological assessment scores using size of contribution (SOC) computed with the following method. We estimated the intercept of the group ('Active' and 'Nonactive') and slope of the neuropsychological assessment scores for subjecting the MEG spectral parameters using the linear regression models which were equivalent to the ANOCOVA model;

$$y = (\beta_0 + \beta_{0i}) + (\beta_1 + \beta_{1i})\,x + \varepsilon \tag{2}$$

where $y$ is one of the MEG spectral parameters (i.e., MF, IAF, or SSE; dependent variable), $\beta_0$/ $\beta_1$ represent the global intercept/ slope across all groups, and $\beta_{0i}$/ $\beta_{1i}$ represent the intercept/ slope for group $i$ (1 = 'Active' or −1 = 'Nonactive'; independent variable). $x$ represents the covariance in the ANCOVA model, which was age or one of the neuropsychological assessment scores (i.e., MMSE-J, FAB-J, or ADAS-J cog). $\varepsilon$ represents a residual of the measurement. For each pair of MEG spectral parameter and covariate (i.e., age, MMSE-J, FAB-J, or ADAS-J cog), the parameters were estimated using the least square method. To enhance the comprehensibility of the results, each covariate was centred (i.e., the mean was subtracted) before model estimation, so that the intercepts represent the estimated $y$ (i.e., MEG spectral parameters; dependent variable) at an average covariate. We defined the SOC as the size of group intercept ($\beta_{0i}$) relative to the global slope ($\beta_1$); therefore, the SOC was computed by dividing ($\beta_{0\,Active}$ - $\beta_{0\,Nonactive}$) by $\beta_1$ for each model, indicating the difference in MEG spectral parameters between groups ('Active' and 'Nonactive') expressed by the unit of neuropsychological assessment scores. The SOC was computed for the pairs of MEG spectral parameters and neuropsychological assessment scores for which the corresponding ANCOVA showed a significant main effect of group (i.e., 'Active' and 'Nonactive') and covariate (i.e., MMSE-J, FAB-J, or ADAS-J cog). A schematic description of our statistical model and SOC is shown in Fig 2. Of note, we did not evaluate nor compare the performances of the regression models. They were designed equivalently to the ANCOVA models in which the terms for group and neuropsychological assessment scores were significant, and the coefficients were solely estimated to supplement the results with SOCs. Therefore, we did not aim to explore the best models or predictors in the present study.

## Results

### Group-level differences and correlations

One hundred forty-seven (79 women and 68 men) and 180 patients (109 women and 71) belonged to the Active and Nonactive groups, respectively. Descriptive statistics and the results of group-level comparison are shown in Table 1. The $t$-tests did not reveal any significant differences in age [$p$ (FDR) = 0.465]. However, the MMSE-J and FAB-J scores were significantly lower [$p$ (FDR) < 0.001], and the ADAS-J cog score was significantly higher [$p$ (FDR) < 0.001] in the Nonactive group than in the Active group. All MEG spectral parameters (MF, IAF, and SSE) were significantly lower [$p$ (FDR) <= 0.001] in the Nonactive group than in the Active group. The results of exploratory correlation analyses are shown in S1 and S2 Tables. Age was negatively correlated with MMSE-J and FAB-J scores in both groups, whereas it was only

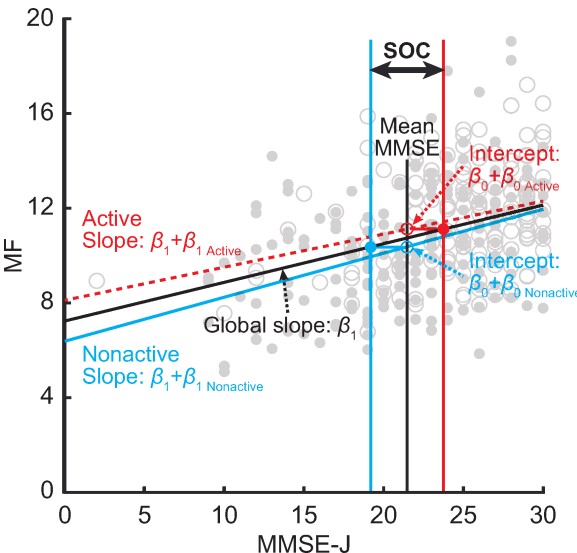

**Fig 2. Schematic Description of the Statistical Model and SOC.** The red dashed and blue solid lines represent the least square lines for the Active and Nonactive groups, respectively. The black vertical line indicates the mean of the neuropsychological assessment score (i.e., MMSE-J) across all patients, whose intersections with the least square lines corresponded to the group intercepts ($\beta_0 + \beta_{0\ Active}$ and $\beta_0 + \beta_{0\ Nonactive}$). The intercepts are converted to the unit of neuropsychological assessment score by horizontally projecting them on the global least square line with a global slope of $\beta_1$, whose gaps between groups are considered as the SOC, representing the effect of group on MF in the unit of MMSE-J at their mean. MF: Median Frequency, MMSE-J, Japanese version of Mini-Mental State Examination; SOC, Size of contribution.

**Table 1. Profile of the dataset and results of group-level comparisons.**

|  | Active | | | Nonactive | | | Two-sample *t*-test | | | |
|---|---|---|---|---|---|---|---|---|---|---|
|  | N | M | SD | N | M | SD | T | df | p (FDR) | d |
| Age | 147 | 77.58 | 0.56 | 180 | 78.16 | 0.55 | −0.73 | 325 | 0.465 | −0.08 |
| MMSE-J | 146 | 24.56 | 0.37 | 180 | 21.54 | 0.40 | 5.41 | 324 | < 0.001* | 0.60 |
| FAB-J | 144 | 11.24 | 0.25 | 179 | 9.73 | 0.25 | 4.22 | 321 | < 0.001* | 0.47 |
| ADAS-J cog | 127 | 12.95 | 0.61 | 136 | 16.20 | 0.76 | −3.32 | 261 | < 0.001* | −0.41 |
| MF | 147 | 11.54 | 0.18 | 180 | 10.39 | 0.19 | 4.33 | 325 | < 0.001* | 0.48 |
| IAF | 147 | 8.85 | 0.06 | 180 | 8.44 | 0.07 | 4.06 | 325 | < 0.001* | 0.45 |
| SSE | 147 | 0.798 | 0.003 | 180 | 0.785 | 0.003 | 3.54 | 325 | 0.001* | 0.39 |

MMSE-J, Japanese version of Mini-Mental State Examination; FAB-J, Japanese version of Frontal Assessment Battery; ADAS-J cog, Japanese version of Alzheimer's Disease Assessment Scale-Cognitive section; MF, Median Frequency; IAF, Individual Alpha Frequency; SSE, Shannon's Spectral Entropy; N, number of patients; *M*, mean; *SD*, standard deviation; *T*, *t*-value; *df*, degree of freedom; *p* (FDR), *p*-value corrected for false discovery rate; *d*, Cohen's *d*.

correlated negatively with ADAS-J cog scores in the Nonactive group and not correlated with MEG spectral parameters in either group. All pairs within neuropsychological assessment scores (i.e., MMSE-J×FAB-J, MMSE-J×ADAS-J cog, and FAB-J×ADAS-J cog) and MEG spectral parameters (i.e., MF×IAF, MF×SSE, and IAF×SSE) were significantly correlated. All pairs between neuropsychological assessment scores and MEG spectral parameters (i.e., MMSE-J×MF, MMSE-J×IAF, MMSE-J×SSE, FAB-J×MF, FAB-J×IAF, FAB-J×SSE, ADAS-J cog×MF, ADAS-J cog×IAF, and ADAS-J cog×SSE) were significantly correlated in both groups, except for the pairs including SSE (i.e., MMSE-J×SSE, FAB-J×SSE, and ADAS-J cog×SSE) for the Active group.

## Effect of groups on MEG spectral parameters

Table 2 summarises the results of the ANCOVA, in which the effect of groups on MEG spectral parameters was investigated while considering the effects of neuropsychological assessment scores. When age was considered as a covariate, the main effect of group (i.e., Active vs. Nonactive) was significant but the covariate and their interaction was not significant for all MEG spectral parameters [Table 2 (A)]. When each neuropsychological assessment (MMSE-J, FAB-J, or ADAS-J cog) was used as a covariate, both the main effects of the group (i.e., Active vs. Nonactive) and covariate were significant for all MEG spectral parameters, while their interactions were not [Table 2 (B)-(D)]. The impact of regular physical activity (i.e., effect of group) on the MEG spectral parameters was quantified as SOC. The SOCs were computed for the MMSE-J, FAB-J, and ADAS-J cog but not for age because the ANCOVA results indicated that age did not contribute to the regression model for any MEG spectral parameters. In the unit of MMSE-J, the SOCs of the groups were 4.14, 3.25, and 11.14 on MF, IAF, and SSE, respectively, indicating that regular physical activity affected the MEG spectral parameters by 3.25–11.14 in terms of MMSE-J scores. Similarly, the SOCs on MF, IAF, and SSE were 3.67, 2.99, and 6.85 in terms of FAB-J, and −10.16, −8.88, and −14.05 in terms of ADAS-J cog scores. The relationships between MEG spectral parameters, neuropsychological assessment scores, and groups are visualised in Fig 3.

## Discussion

This study revealed that MEG spectral parameters (i.e., electrophysiological brain activity) are influenced by regular physical activity as well as neuropsychological assessment scores (i.e., current cognitive states) (Tables 1 and 2). The influence of regular physical activity on MEG spectral parameters was significant even when controlling for the neuropsychological scores (Fig 2 and Table 2).

Recent studies have revealed that the prevalence of dementia is declining, at least in some countries [71,72]. It is considered that this decline is related to changes in lifestyle. Several factors associated with a 'healthy lifestyle' contribute to reducing the risk of cognitive impairment [7] and these factors interfere with each other. In the present study, we focused on regular physical activity as a representative index of lifestyle-associated factors. It reduces the risk of

**Table 2. Assessing the effects of regular physical activity on MEG spectral parameters.**

| | | Dependent variable | | | | | | | | |
| | | MF | | | IAF | | | SSE | | |
| | Independent variable | $F$ | $p$ | $\eta^2$ | $F$ | $p$ | $\eta^2$ | $F$ | $p$ | $\eta^2$ |
|---|---|---|---|---|---|---|---|---|---|---|
| (A) | Age | 0.98 | 0.323 | 0.003 | 1.94 | 0.165 | 0.006 | 0.52 | 0.471 | 0.002 |
| | RPA | 18.38 | < 0.001* | 0.054 | 15.99 | < 0.001* | 0.047 | 12.26 | 0.001* | 0.036 |
| | Age × RPA | 0.45 | 0.505 | 0.001 | 0.03 | 0.860 | 0.000 | 1.01 | 0.317 | 0.003 |
| (B) | MMSE-J | 45.99 | < 0.001* | 0.123 | 52.15 | < 0.001* | 0.138 | 6.87 | 0.009* | 0.020 |
| | RPA | 6.07 | 0.014* | 0.016 | 4.39 | 0.037* | 0.012 | 6.82 | 0.009* | 0.020 |
| | MMSE-J × RPA | 0.79 | 0.375 | 0.002 | 0.44 | 0.510 | 0.001 | 0.16 | 0.694 | < 0.001 |
| (C) | FAB-J | 39.29 | < 0.001* | 0.106 | 44.15 | < 0.001* | 0.119 | 10.10 | 0.002* | 0.030 |
| | RPA | 9.56 | 0.002* | 0.026 | 7.76 | 0.006* | 0.021 | 7.52 | 0.006* | 0.022 |
| | FAB-J × RPA | 3.12 | 0.078 | 0.008 | 1.33 | 0.250 | 0.004 | 2.66 | 0.104 | 0.008 |
| (D) | ADAS-J cog | 35.85 | < 0.001* | 0.116 | 38.44 | < 0.001* | 0.125 | 7.62 | 0.006* | 0.028 |
| | RPA | 13.43 | < 0.001* | 0.044 | 10.99 | 0.001* | 0.036 | 5.74 | 0.017* | 0.021 |
| | ADAS-J cog: × RPA | 0.20 | 0.659 | 0.001 | 0.21 | 0.650 | 0.001 | < 0.01 | 0.973 | < 0.001 |

RPA, regular physical activity; MMSE-J, Japanese version of Mini-Mental State Examination; FAB-J, Japanese version of Frontal Assessment Battery; ADAS-J cog, Japanese version of Alzheimer's Disease Assessment Scale-Cognitive section; RPA, regular physical activity; MF, Median Frequency; IAF, Individual Alpha Frequency; SSE, Shannon's Spectral Entropy; $F$, $F$-statistic; $p$, $p$-value.

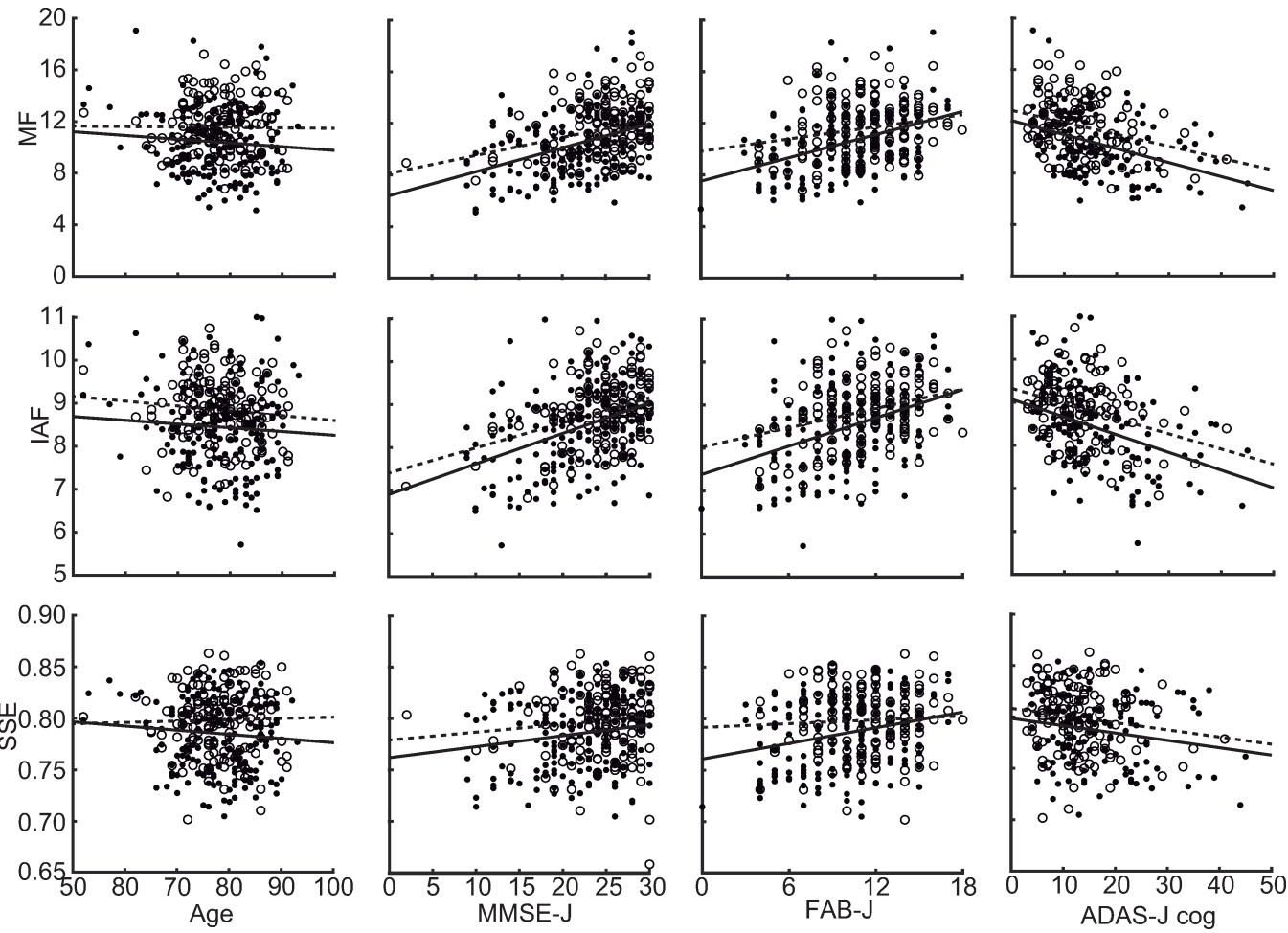

**Fig 3. Scatterplots of the Relationships between MEG Spectral Parameters and Age/ neuropsychological Assessment Scores.** Open dots and filled dots represent Active and Nonactive groups, respectively. Dashed and solid lines represent least square lines for Active and Nonactive groups, respectively. MF, median frequency; IAF, Individual alpha frequency; SSE, Shannon's spectral entropy; MMSE-J, Japanese version of Mini-Mental State Examination; FAB-J, Japanese version of Frontal Assessment Battery; ADAS-J cog, Japanese version of Alzheimer's Disease Assessment Scale-Cognitive.

cognitive impairment both directly and indirectly [7,73]. As direct pathways, physical activity improves blood flow, reduces inflammation, induces the synthesis and release of neurotrophins, and modulates of amyloid β turnover, which improve cognition [74]. As indirect pathways, regular physical activity reduces other risk factors such as hypertension [75], obesity [76], diabetes [77]. Lifestyle-associated factors synergically reduce the risk of cognitive impairment and it is impractical to identify individual beneficial effects of regular physical activity. From a clinical perspective, it is not clear whether regular physical activity reduces the risk of cognitive decline and maintains the current cognitive state directly or indirectly. Thus, we considered these factors as a group and focused on the comprehensive effects of regular physical activity, which are of interest in clinical practice.

This study showed that MEG spectral parameters were associated with regular physical activity and correlated with neuropsychological assessment scores at the group-level (Fig 3, Table 1 and S1 and S2 Tables), which supported our hypothesis. These results suggest that MEG spectral parameters reflect two independent aspects: (1) current cognitive states (i.e., neuropsychological assessment scores) and (2) additional changes in electrophysiological brain activity

associated with physical activity. To quantify the additional changes, we introduced SOC which indicated that the additional increases in MEG spectral parameters driven by the group (i.e., influence of regular physical activity) are equivalent to the neuropsychological scores of 3.25–11.14 in MMSE-J, 2.99–6.85 in FAB-J, and 8.88–14.55 in ADAS-J cog. According to the current cognitive state, correlation analyses showed that relationships between MEG spectral parameters and neuropsychological scores are differently modulated by the groups (Active and Nonactive) (S1 and S2 Tables). In the Nonactive group, all MEG parameters (i.e., MF, IAF, and SSE) were correlated with all neuropsychological parameters (i.e., MMSE-J, FAB-J, and ADAS-J cog), while in the Active group, only MF and IAF were correlated with all neuropsychological parameters, and SSE was correlated with none of the three neuropsychological parameters (S1 and S2 Tables). These findings suggest that regular physical activity differently affects SSE and MF/IAF. The MF/IAF represent frequency power balance between low and high-frequency oscillatory activities, whereas SSE captures the complexity; a unique aspect of the spectral components contained in electrophysiological activity. As the patients with cognitive impairment lose the complexity, the SSE decreases as symptoms progress [25], suggesting positive correlations between SSE and neuropsychological parameters. Our results showed that regular physical activity biased this relationship, indicating that patients in the Active group maintained comparable complexity in their electrophysiological oscillatory components even when their cognitive state was impaired. This implies that regular physical activity had an additional influence on the electrophysiological activity in the Active group. Taken together, it is plausible that regular physical activity modified the electrophysiological activity (i.e., MEG spectral parameters). The influence of regular physical activity was evident in all MEG spectral parameters, while it was most salient for the SSE, because the relationships between SSE and current cognitive states (i.e., neuropsychological parameters) were biased by the regular physical activity.

The additional influence reminds us of the 'cognitive reserve' concept, which is defined as an adaptability of cognitive processes that helps to explain differential susceptibility of cognitive abilities or day-to-day function to brain ageing, pathology, or injury [78]. Physical, cognitive, and social activities increase the cognitive reserve and attenuate the effects of neuropathology on cognitive states [79]. Its physiological backgrounds are based on axonogenesis and synaptogenesis [10], greater efficiency of and less decline in functional brain networks [80,81], and neuroplasticity [11]. These physiological changes are induced by physical activity [82–84]. A previous MEG study showed that a high cognitive reserve is associated with enhanced high-frequency oscillatory (gamma) activity [85], which is related to synaptic activities [65], network activities [66], and neuroplasticity [67]. Altogether, we speculate that regular physical activity leads to physiological changes in the brain that facilitate the adaptation to difficulties in daily live for patients with cognitive impairment, and that these changes are associated with high values in MEG spectral parameters, especially in SSE. For the clinical practice, it is promising that MEG spectral parameters can be used to evaluate positive effects of lifestyle-associated factors during medical consultations [34].

## Limitations

The current study has some limitations. First, this study was conducted in a single hospital with an MEG centre. To generalise the present findings, they should be replicated in other hospitals with different patients. We are currently preparing similar studies with various collaborating hospitals. Second, we cannot rule out the possibility of reverse causality. Although we assumed that regular physical activity enhanced the MEG spectral parameters, the causality might in fact be opposite; patients with high MEG spectral parameters could prefer a healthy lifestyle including regular physical activity. There could also be no causal relationship; for example, both could be consequences of other factors, such as different lifestyle-associated, genetic, environmental, and/or unknown factors. We cannot exclude these possibilities because this was a retrospective observational study, not an intervention study. However, the present findings are still valuable for the clinical practice. There is an association between regular physical activity and MEG spectral parameters which allows physicians to infer patients' lifestyles. Patients with lower MEG spectral parameters than other patients with comparable current cognitive states will likely not have healthy lifestyles and should be encouraged to change them. We have previously

reported the efficacy of this strategy in a case series [34] which was an intervention study. Future longitudinal studies are required to clarify the causality. Third, some potential confounding factors were not considered in the present study, both lifestyle-associated (e.g., education and environment) [86,87] and other factors (e.g., genetic factors) [88]. We have examined the comprehensive effects of regular physical activity, which includes its interactions with other lifestyle-associated factors. However, some lifestyle-associated and/or other factors could influence MEG spectral parameters independently from regular physical activity, which should have been controlled for. However, because this was a retrospective observational study based on existing clinical records, only limited information was available. We are planning further studies to investigate the influence of other factors.

## Conclusions

MEG spectral parameters capture additional influences of regular physical activity on brain activity as well as the current state of cognition (i.e., neuropsychological assessment scores). These measurements can assist physicians in designing therapeutic strategies and promoting lifestyle changes to prevent cognitive decline.

## Supporting information

**S1 Table. Results of the Correlation Analysis: Active Group.** MMSE-J, Japanese version of Mini-Mental State Examination; FAB-J, Japanese version of Frontal Assessment Battery; ADAS-J cog, Japanese version of Alzheimer's Disease Assessment Scale-Cognitive section; MF, Median Frequency; IAF, Individual Alpha Frequency; SSE, Shannon's Spectral Entropy; r, Pearson's correlation coefficient; p (FDR), p-value corrected for false discovery rate.
(PDF)

**S2 Table. Results of the Correlation Analysis: Nonactive Group.** MMSE-J, Japanese version of Mini-Mental State Examination; FAB-J, Japanese version of Frontal Assessment Battery; ADAS-J cog, Japanese version of Alzheimer's Disease Assessment Scale-Cognitive section; MF, Median Frequency; IAF, Individual Alpha Frequency; SSE, Shannon's Spectral Entropy; r, Pearson's correlation coefficient; p (FDR), p-value corrected for false discovery rate.
(PDF)

## Acknowledgments

We express our sincere gratitude to the patients who participated in this study. We would also like to affirm our genuine respect for their contributions to the continued progress of medical sciences. We sincerely thank Dr. Hajime Kamada (Honorary chairperson, Hokuto Hospital) and Dr. Ikuo Hashimoto (Chairperson, Kumagaya General Hospital) for providing access to the facilities. We would also like to thank Editage (www.editage.com) for the English language editing.

## Author contributions

**Conceptualization:** Keisuke Fukasawa, Hideyuki Hoshi, Yoshihito Shigihara.

**Data curation:** Keisuke Fukasawa, Yoko Hirata, Momoko Kobayashi, Keita Shibamiya, Sayuri Ichikawa, Yoshihito Shigihara.

**Formal analysis:** Hideyuki Hoshi, Yoshihito Shigihara.

**Funding acquisition:** Yoshihito Shigihara.

**Investigation:** Yoshihito Shigihara.

**Methodology:** Hideyuki Hoshi, Yoshihito Shigihara.

**Project administration:** Yoshihito Shigihara.

**Resources:** Yoshihito Shigihara.

**Software:** Yoshihito Shigihara.

**Supervision:** Yoko Hirata.

**Validation:** Keisuke Fukasawa, Yoshihito Shigihara.

**Visualization:** Yoshihito Shigihara.

**Writing – original draft:** Hideyuki Hoshi, Yoshihito Shigihara.

**Writing – review & editing:** Hideyuki Hoshi, Yoshihito Shigihara.

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
