## [Decision Letter · Decision Letter 0]

PONE-D-24-41991Regular physical activity affects brain activities in old individuals: an observational studyPLOS ONE

Dear Dr. Shigihara,

Thank you for submitting your manuscript to PLOS ONE. After careful consideration, we feel that it has merit but does not fully meet PLOS ONE’s publication criteria as it currently stands. Therefore, we invite you to submit a revised version of the manuscript that addresses the points raised during the review process.

I received contrasting opion from reviewers. The prevalent opinion, including my own, is tat tha manuscript needs substantial revisions. Criticism, however, mainly affects the description (completeness, clarity) of methods and statistical teratment. This is within the limits of a major revision. Please address all the comments by reviewers regarding methodology and also other aspects, including coherence bethween results and interpretation.

We look forward to receiving your revised manuscript.

Kind regards,

Federico Giove, PhD

Academic Editor

PLOS ONE

Journal requirements: When submitting your revision, we need you to address these additional requirements. 1. Please ensure that your manuscript meets PLOS ONE's style requirements, including those for file naming. The PLOS ONE style templates can be found at https://journals.plos.org/plosone/s/file?id=wjVg/PLOSOne_formatting_sample_main_body.pdf and https://journals.plos.org/plosone/s/file?id=ba62/PLOSOne_formatting_sample_title_authors_affiliations.pdf. 2. Thank you for stating the following financial disclosure:  [This study was partially supported by RICOH Co., Ltd].  Please state what role the funders took in the study.  If the funders had no role, please state: ""The funders had no role in study design, data collection and analysis, decision to publish, or preparation of the manuscript."" If this statement is not correct you must amend it as needed. Please include this amended Role of Funder statement in your cover letter; we will change the online submission form on your behalf. 3. Please note that your Data Availability Statement is currently missing [the repository name and/or the DOI/accession number of each dataset OR a direct link to access each database]. If your manuscript is accepted for publication, you will be asked to provide these details on a very short timeline. We therefore suggest that you provide this information now, though we will not hold up the peer review process if you are unable.  4. Your ethics statement should only appear in the Methods section of your manuscript. If your ethics statement is written in any section besides the Methods, please delete it from any other section.  5. We are unable to open your figures [figure1_psy_meg_20240823d.eps and figure2_interpretation_20241003a.eps]. Please kindly revise as necessary and re-upload.

Reviewers' comments:

Reviewer's Responses to Questions

**Comments to the Author**

1. Is the manuscript technically sound, and do the data support the conclusions?

Reviewer #1: Yes

Reviewer #2: Yes

Reviewer #3: Yes

Reviewer #4: Partly

Reviewer #5: No

2. Has the statistical analysis been performed appropriately and rigorously? 

Reviewer #1: Yes

Reviewer #2: No

Reviewer #3: Yes

Reviewer #4: Yes

Reviewer #5: No

3. Have the authors made all data underlying the findings in their manuscript fully available?

Reviewer #1: No

Reviewer #2: Yes

Reviewer #3: Yes

Reviewer #4: No

Reviewer #5: Yes

4. Is the manuscript presented in an intelligible fashion and written in standard English?

Reviewer #1: Yes

Reviewer #2: Yes

Reviewer #3: Yes

Reviewer #4: Yes

Reviewer #5: Yes

5. Review Comments to the Author

Reviewer #1: The manuscript described whether the repeated physical activities affect the aged people’s brain function or not. The authors divided the 327 patients in memory clinic into activity group and non-activity group, and measured the brain function using MEG as well as psychological tests. They found the promising results by repeated activity improving the cognitive functions as well as the good outcome in the future.

There are several problems in this manuscript.

1) The design of the study is not prospective, just retrospective transverse research. Therefore, there is a problem suggesting the reverse results; the high cognitive function people tend to perform more physical activities. Furthermore, this study was done in one institute, then the bias of the subjects could not be avoided. One suggested plan is that the repeated research on different subjects might be more stronger influences on this result.

2) Confounding factors such as diet, social activities, educational level may affect the cognitive functions.

3) In my experience, the relationship between MEG results and cognitive function is considered to be difficult to understand.

4) Data availability is not clearly written. DOI is not effective.

Reviewer #2: The paper statistically evaluates correlation of personal physical activity status, cognitive level and values of brain activity of large old-age population.

Compliance with criteria of PLOS ONE.

The study presents the results of primary scientific research – Yes

Results reported have not been published elsewhere – I trust the authors.

Experiments, statistics, and other analyses are performed to a high technical standard and are described in sufficient detail – Not really, see my detailed comments below.

Conclusions are presented in an appropriate fashion and are supported by the data – So, so.

The article is presented in an intelligible fashion and is written in standard English – The authors used a specialized profiling firm to edit their English text (see l. 326-327).

The research meets all applicable standards for the ethics of experimentation and research integrity – Yes

The article adheres to appropriate reporting guidelines and community standards for data availability – Yes. See l. 348-350.

Detailed comments and notes

1) Can the authors provide references to explanations of values MF, IAF, SSE (l. 84).

2) Abbreviations in main text differ from those in the abstract (34-38) For example, in the abstract you note MMSE-J, but in the text just MMSE (53).

3) 7, 111 – give corresponding statistics for men, especially statistics of age.

4) 134-140. You here mention about special Japanese variant of corresponding scores. I think, you should provide concise description of differences them from international. In this context you should change abbreviations in the abstract.

5) 153 – abbreviation “meg” must be in uppercase.

6) In Methods you should describe hardware used for MEG abduction, protocol of this procedure, bands and channels you used for future analysis. The paper has complete lack of this technical information.

7) I suppose a good idea would to provide short mathematical description of all parameters you analyze (MF, IAF, SSE).

8) 160-165. Provide mathematical formula for SOC value.

9) 173 – abbreviation “meg” must be in uppercase.

10) 171 – 174. You write about significant differences in values. I think, you may provide here statistical results (p=…, etc.) mentioning the statistical method of evaluation.

11) 181. Please provide more informative header of the table. What is P(unk)? In the text above (l. 173-179) you mention Spearman’s rank correlation but the table doesn’t provide any information about it. By the way the Table 1 provides information about groups of patients. I think you should include one more table with result of Spearman’s rank correlation analysis.

12) Again, 188 - abbreviation “meg” must be in uppercase. Please look through the text and correct this error.

13) 188-191. Excuse me but I see no strong correlation between age and scores. There mostly are horizontal linear regression lines. I am strongly against “ritual” phrases in statistical reports (l. 191). If you mention ANOVA you should describe statistical model, you used, like one-way, two-way, repeated measurement and so on. Statistical results should have F and p values and information about post-hoc estimation of pair-wise comparison.

14) 202-209. From the text and table, I deduce that you use two-way ANOVA with two main factors 1) MEG value and 2) RPA to assess their influence on score parameter. I think, you should describe it in the text. Why do you not analyze the group of activity as impact factor?

15) 210. Title of the table is bad. It describes statistical method. I think the title could give the reader idea what is estimated by the table. I would propose something from your text “Assessing the effects of regular physical activity on MEG spectral parameters”. And statistical method you could describe somewhere in Table note below.

16) Better idea to introduce and describe Fig 2 in Results.

Reviewer #3: This manuscript is of interest for the general audience of PLOS One. I believe that it is well structured and reads very well. I have a few comments (please see below) which I hope that the authors can clarify in a revised version.

Can the authors provide more details on the data acquisition (experimental paradigm and recordings= specifications) as well as the preprocessing (e.g., filtering) of the MEG data? The manuscript does not contain any information on this apart from descriptions on the three spectral outcomes.

The authors write that patients were classified as either ‘active’ or ‘nonactive’. The definition seems quite vague as it seems that patients reported these exercise levels themselves. Was this standardised in any way? For example, can the authors be more precise on what was meant with being physically active? Did the patients need to meet any criteria to be labelled active (e.g., in terms of frequency or duration)? Similarly, is it possible to have a more precise definition on what is meant with being 'nonactive'? If this is the most detailed information that the authors can provide, how sure are they that the ‘active’ and ‘nonactive’ groups are defined correctly?

Minor comments

Line 154: meg -> MEG

Line 173: meg -> MEG

Line 171 and 181: How is it possible that the p-values are different between the in-text report (line 171; p = 0.647) and Table 1 (line 181; p = 0.305)?

Line 181 (Table 1): Can the authors please report test statistics, degrees of freedom and effect sizes for these statistical contrasts?

Line 188: meg -> MEG

Reviewer #4: Fukasawa et al. conducted a retrospective study on a large group of patients referred to their dementia clinic. They categorized the patients into active and inactive groups based on their routine physical activity and compared their cognitive scores as well as MEG spectral parameters. Their findings suggested that patients who engaged in regular physical activity tended to have higher cognitive scores.

While I read the submitted manuscript with interest, it requires significant revisions to improve clarity, coherence, and scientific rigor. Below are detailed suggestions for each section of the manuscript:

Introduction

o The purpose of the first paragraph is unclear. Is it intended to define dementia, explain why specific tests were used, or discuss the difficulty of predicting cognitive decline in dementia patients? Clarify the focus.

o The opening sentence of the second paragraph makes a broad claim but lacks references to support it. Provide appropriate citations.

o The concept of "functional reserve" should be explained before introducing it.

o This paragraph appears to discuss the role of physical activity, but the connection to cognitive reserve is unclear. Use examples that directly relate to cognitive reserve (e.g., mental stimulation, education) rather than unrelated analogies like kidney/liver functions.

o Explain the physiological mechanisms by which physical activity might prevent cognitive decline, with appropriate references to support these claims.

o In the third paragraph the claim that electrophysiological measurements do not correlate well with cognitive assessment may reflect issues with the measurements themselves, not necessarily the impact of cognitive reserve due to physical activity. Avoid oversimplified assumptions and provide a robust explanation.

o Explain how spectral measurements are proposed to demonstrate cognitive reserve. Clarify why cognitive reserve might not directly improve cognitive performance but could indirectly affect spectral measures.

o The hypothesis that electrophysiological activity represents both functional reserve (via physical activity) and cognitive state independently is unclear. Provide a logical connection between these concepts and how they are tested in this study.

o Overall, the introduction would benefit from a more coherent structure, improved clarity, and a less exaggerated presentation of the hypothesis.

Methods

1. Dataset:

o Define the threshold used to classify participants as cognitively healthy.

2. Parameters:

o Avoid claiming that "these parameters are established in clinical practice" if this assertion is based on a single study, and a case serios, particularly if conducted by the same group. Broader validation is necessary to support this statement.

o Define "regular exercise routine" explicitly. Was this based on general responses (e.g., "Do you exercise regularly?") or specific criteria (e.g., "Do you walk 30 minutes at least three times per week?"). If the former, this grouping may lack reliability.

3. Statistics:

o Justify the use of Wilcoxon tests. For a dataset of this size, a t-test or ANOVA would be more appropriate and straightforward.

4. MEG analysis:

o There were no mention of how the analysis of MEG data have been done, the preprocessing of the data as well as the calculation of spectral parameters should be explained in details.Results

1. Correlations:

o The manuscript reports significant correlations between cognitive scores and cognitive measures, as well as between MEG measures and MEG measures, but fails to explore correlations across these domains (e.g., MEG measures with cognitive scores). This should be addressed.

2. Evaluation of Regression Models and Physical Activity Effect:

o These two sections are unclear to me. Clarify the methods and results to make the findings interpretable.

Discussion:

• The statement that "[MEG spectral parameters] indicate a functional reserve that may help prevent future cognitive decline, regardless of the current cognitive state" is not supported by the findings. How was cognitive reserve measured in this study?

• The conclusions drawn from the results are overgeneralized and not sufficiently supported. Avoid overinterpreting simple correlations.

• The discussion is overly focused on functional reserve, a concept that appears unrelated and untested within the study. This weakens the narrative and risks misrepresentation of the findings.

General Comments

• To demonstrate that physical activity prevents cognitive decline, the study should begin by establishing a correlation between the amount of physical activity and cognitive scores or show a significant difference in cognitive scores between active and inactive groups. The relevance of MEG measurements is unclear in this context.

• To support the claim that physical activity prevents cognitive decline, the manuscript should first establish a direct correlation between physical activity and cognitive scores. Alternatively, compare cognitive scores between physically active and inactive groups.

• The inclusion of MEG measurements seems disconnected from the primary research question. If MEG is to be used, consider using logistic regression to predict activity status (active vs. inactive) based on MEG and cognitive measures. Then, test whether including MEG measurements improves predictive power.

• Simplify statistical analyses. The use of overly complex models for simple comparisons detracts from clarity and accessibility.

Conclusion

The manuscript has potential but requires substantial revisions to address the issues outlined above. The introduction needs a more focused narrative, the methods and statistical analyses need better justification, and the results should be clarified and aligned with the hypotheses. Furthermore, the conclusions should be restrained to reflect the actual findings without overstating their implications.

Reviewer #5: In this study, Fukasawa and colleagues investigated whether electrophysiological brain activity can reflect the preventative effect of regular physical activity against cognitive decline. The authors combine records of regular physical activity, with neurophysiological recordings obtained using Magnetoencephalography (MEG) and neuropsychological assessments of cognitive function. Regression analyses were used to investigate the effect of physical activity and age on the three spectral MEG parameters. The study also aimed to disentangle the contribution of physical activity to brain signals and neuropsychological assessments. While I commend the authors for investigating such an interesting and complex phenomenon, there are serious issues that prevent me from recommending publication of this work. The following is a summary of the most important issues:

Major issues:

1. The manuscript lacks any kind of detail about how MEG data was collected, preprocessed and analyzed. Without such information, it is impossible to establish whether MEG data has been treated according to scientific conventions and whether the MEG analysis pipeline could have influenced the current results. Because MEG spectral measures are a central part of the current study, the authors should have included a detailed subsection to within their methods section explaining each step in the MEG data analysis pipeline. Such section should include, among other things: the MEG system used, the manufacturer, whether online filters were applied, what instructions were given to participants, whether data was acquired in eyes-closed or eyes-open resting-state conditions, what package was used to analyze MEG data, off-line filters, procedures to identify bad channels or noisy segments, whether these were manual or automatized, whether data was epoched or not, and if so, what was the epoch’s duration, whether epochs were rejected, and if so, what epoch rejection thresholds or parameters were used, the total number of epochs before and after epoch rejections per individual, among others.

Similarly, regarding MEG data analysis, the authors need to provide a detailed description of how spectral parameters were obtained, describing all relevant steps, methods, tools used in their signal-processing pipeline and mathematical formulas (if necessary). For instance, it is not clear what type of method was used to compute the spectral representation of the time-domain signal, what parameters were set for such analysis, and how the spectral measures were subsequently estimated. An example of how to report both MEG preprocessing and spectral analyses can be found in references 20 and 23 of the manuscript, or in other similar papers by that same group.

2. There are serious issues with how statistical analyses are reported. For instance, the authors report their correlation analyses and direct the reader to table 1 [line 179], but I believe table 1 actually presents the results of the Wilcoxon Signed Rank Tests. Relatedly, the authors state that Fig 1 shows the regression analyses [line 188], but the title of Fig1 indicates these scatter plots represent the correlation between spectral parameters and neuropsychological assessment. It is therefore unclear what analysis figure 1 actually shows.

The regression analyses must be described and reported in full detail, but instead, the authors provide almost no information about the results of their regression analyses in their result section, other than referring the reader to figure 1. First of all, authors need to explicitly state how models were specified (provide their formulas). For instance, the author’s state:

“We then assessed the effect of regular physical activity on meg spectral parameters while accounting for cognitive state. Each meg spectral parameter (MF, IAF, and SSE) was subjected to a multiple regression model with three predictors: (1) a categorical predictor of regular physical activities (‘Active’ and ‘Nonactive’), (2) age or one of the neuropsychological assessment scores (MMSE-J, FAB-J, or ADAS-J cog) as a covariate, and (3) their interaction term.” [Lines 152-157].

Based on this, I would assume that the models took the form:

y ~ RPA + AGE + RPA*AGE

y ~ RPA + MMSE-J + RPA*MMSE-J

y ~ RPA + FAB-J + RPA*FAB-J

y ~ RPA + ADAS-J + RPA*ADAS-J

This is just an assumption, and I do not know if this is correct, which illustrates the problem: there is very little information about how many models were designed and how they were specified. Without this information, it is very difficult to assess whether models were appropriate and whether the author’s interpretation of their results are justified.

Regression tables could be used to summarize the results, illustrating whether the effect of each term within each model is actually significant or not. Alternatively, this information could be presented within the text. Regardless, it is crucial that the reader understands whether factors or their interaction have a significant effect in predicting changes in the dependent variable. Such information, however, is not provided.

3. To properly evaluate model performance, authors should also report metrics of goodness-of-fit for each model, such as R2, AIC, or BIC for each model. Unfortunately, none of these values are reported. The authors report SOC (Size of Contribution) as a measure of how much one factor contributes to model performance, but this is not enough to back up the interpretations they derive from their regression analyses in their discussion section. I am also having difficulties understanding table 2. The authors state that

“To evaluate the regression models assessing the effects of regular physical activity on MEG spectral parameters while accounting for age or one of the neuropsychological assessment scores, we performed an ANOVA (Table 2)”.

Based on this, I would expect a stepwise model comparison procedure where covariates are sequentially added to initial models and then the statistical effect of adding such covariates is assessed via maximum likelihood estimation. Instead, the authors present table 2 as a summary of their model comparison procedure. However, it is difficult to understand from table 2 how models were compared and how the effect of covariates was assessed. Actually, table 2 gives the impression that spectral parameters were treated as predicting (i.e. independent) variables because they are reported with associated F and p values, as if they were factors within a new set of independent models. It is therefore not clear how different models were compared, or how the presence or absence of their covariates of interest affected their models.

4. A somehow worrisome issue at the conceptual level is the use of ambiguous terminology in the introduction. Throughout this section, the authors use two concepts, functional reserve and cognitive reserve, which are not mutually interchangeable. Functional reserve is physiologically quantifiable, while cognitive reserve is behaviorally, rather than physiologically quantifiable. I understand the authors attempt to establish a parallelism between functional and cognitive reserve. However, in later sections of the manuscript, it remains unclear whether the authors are empirically concerned with functional reserve (which might be quantifiable via MEG), cognitive reserve (which would be quantifiable via neuropsychological assessment), or both. What is more, the authors later state:

“We hypothesized that electrophysiological brain activity would represent both the amount of functional reserve (i.e., regular physical activity) and the current cognitive state independently” .

This implies that regular physical activity is their operationalization of functional reserve. Such a lack of conceptual clarity makes it difficult to connect relevant concepts with their operationalized independent variables of interest. Along with the aforementioned issues in how regression models are specified and reported, it is very difficult to assess the validity of the interpretations offered by authors in the discussion section of their work.

Other issues

1. It would be useful if the authors offered a more detailed description of “cognitive states”, especially considering the broad scope of the journal. I assume that when they use the term “cognitive state”, they refer to the clinical assessment of multiple cognitive functions such as attention, memory, alertness, among others, as quantified by clinically-validated neuropsychological questionnaires and assessment tools.

2. The authors are right that MEG measures the magnetic fields produced by the electrophysiological activity of populations of neurons in the cerebral cortex, but it is inaccurate to state that MEG reflects dementia-associated cognitive states [Lines 79-81]. MEG cannot directly reflect cognitive states, but rather, provide measures of brain function that might (or might not) be indicative of those states. Relatedly, while the authors do well in justifying their three spectral measures of interest, they should explicitly acknowledge that these here are just a few among many more parameters and measures of brain function that can be obtained using MEG.

3. The authors state that “This study revealed that regular physical activity significantly influenced electrophysiological brain activity while controlling for the current cognitive state (i.e., neuropsychological scores). This influence was significant across any neuropsychological assessment scores considered”, but considering the lack of clarity on MEG and statistical analysis, it is hard to see how they arrived at this conclusion. This might also be an overinterpretation of their results, particularly because MEG-measured brain dynamics during resting state are known to be influenced by a myriad of genetic and environmental factors, which are not controlled within the current experimental design.

4. In the same vein, an important covariate that the authors did not include in their analyses was educational attainment. This is important because the authors cite previous research linking educational attainment with cognitive reserve (Stern et al. 1992), and previous studies investigating the effect of physical activity on MEG-measured brain activity have in fact included educational attainment as a control variable (de Frutos-Lucas et al. 2018).

de Frutos-Lucas, J., López-Sanz, D., Zuluaga, P., Rodríguez-Rojo, I. C., Luna, R., López, M. E., Delgado-Losada, M. L., Marcos, A., Barabash, A., López-Higes, R., Maestú, F., & Fernández, A. (2018). Physical activity effects on the individual alpha peak frequency of older adults with and without genetic risk factors for Alzheimer’s Disease: A MEG study. Clinical Neurophysiology, 129(9), 1981–1989. https://doi.org/10.1016/j.clinph.2018.06.026

Stern Y, Alexander GE, Prohovnik I, Mayeux R. Inverse relationship between education and

402 parietotemporal perfusion deficit in Alzheimer’s disease. Ann Neurol. 1992 Sep;32(3):371–5.

5. The last paragraph of the discussion section where the authors discuss why some patients in the active group use functional reserve to improve their current cognitive state [lines 287-305] is highly speculative, as claims are not directly related to or backed up by the authors’ results. Particularly, lines 298-305 present an oversimplification of how electrophysiological neural dynamics are generated and linked to brain disease. Many of these claims need to be toned down.

6. PLOS authors have the option to publish the peer review history of their article (what does this mean? ). If published, this will include your full peer review and any attached files.

**Do you want your identity to be public for this peer review?** For information about this choice, including consent withdrawal, please see our Privacy Policy .

Reviewer #1: No

Reviewer #2: **Yes: ** Andriy Gorkovenko

Reviewer #3: No

Reviewer #4: No

Reviewer #5: No

---

## [Author Response · Author response to Decision Letter 1]

15 Feb 2025

Response to reviewers:

Regular physical activity affects brain activities in old individuals:

an observational study

Short title: Physical activity and brain activity

Keisuke Fukasawa1, ¶, Hideyuki Hoshi2,3, ¶, Yoko Hirata4, Momoko Kobayashi3, Keita Shibamiya3, Sayuri Ichikawa1, Yoshihito Shigihara2,3*

1Clinical Laboratory, Kumagaya General Hospital, Kumagaya City, Saitama, Japan,

2Precision Medicine Centre, Hokuto Hospital, 7-5 Kisen, Inada-cho, Obihiro-shi, Hokkaido, Japan

3Precision Medicine Centre, Kumagaya General Hospital, Kumagaya City, Saitama, Japan

4Department of Neurosurgery, Kumagaya General Hospital, Kumagaya City, Saitama, Japan

* Corresponding author Yoshihito Shigihara

E-mail: y-shigihara@hokuto7.or.jp (YS)

¶These authors contributed equally to this work.

To the editor and reviewers:

Thank you for giving us the opportunity to submit a revised draft of our manuscript titled ‘Regular physical activity affects brain activities in old individuals: an observational study’.

We appreciate the time and effort you have dedicated to providing valuable feedback on our manuscript. We are grateful for your insightful comments that helped us improve our paper's quality. We have incorporated changes to reflect all of your suggestions.

Please find our point-by-point responses to all comments and concerns below.

Role of Funder statement

This study was partially supported by RICOH Co., Ltd. The funders had no role in study design, data collection and analysis, decision to publish, or preparation of the manuscript.

Response to Reviewer 1 Comments

Comment 1-1:

The design of the study is not prospective, just retrospective transverse research. Therefore, there is a problem suggesting the reverse results; the high cognitive function people tend to perform more physical activities. Furthermore, this study was done in one institute, then the bias of the subjects could not be avoided. One suggested plan is that the repeated research on different subjects might be more stronger influences on this result.

Response:

Thank you for your comments and suggestions. Following the comment, we have modified the manuscript in lines 434-437 and 437-450 in the revised manuscript.

Regarding the first point, we fully agree that there is a possibility of reverse results. As mentioned, it is an unavoidable limitation of the present study because it was an observation study. Although we had already mentioned this in the limitation section of the original manuscript, we have further clarified it (pages 26-27, lines 437-450). While the possibility of reverse causality remains, this study provides meaningful insight for the clinical practice. The MEG spectral parameters were different between groups (‘Active’ vs. ‘Nonactive’), indicating that the parameters are good indicators for physicians to infer patients’ lifestyles. We believe that it does not matter if the neurological differences yielded the patients’ active/nonactive behaviour or whether the patients’ active/nonactive lifestyles yielded the neurological differences, because the results could still contribute to properly assessing patients’ conditions and designing effective therapeutic strategies.

Regarding the second point, the claim is on point. We have acknowledged that this is one of the limitations of the present study (page 27, lines 434-437). To address the concern, we are currently preparing to replicate the study in another hospital equipped with MEG. Furthermore, we know that another MEG site in Japan is planning similar studies. We hope that future studies from those MEG sites would provide supportive evidence.

Comment 1-2:

Confounding factors such as diet, social activities, educational level may affect the cognitive functions.

Response:

Thank you for the insightful comment. Accordingly, we have added descriptions regarding the relationships between regular physical activity and other lifestyle-associated (‘confounding’) factors (pages 23-24, lines 370-384) and added a limitation regarding the confounds (pages 27-28, lines 450-458).

Lifestyle-associated factors interact with each other. They synergically modify the cognitive state. For example, a regular exercise routine contributes to a reduced risk of cognitive decline both directly and indirectly. The indirect pathway manifests via changing other risk factors (e.g., reducing the risk of hypertension). From a clinical perspective, it is not important to discuss how much each lifestyle-associated factor directly contributed to (reducing) the risk of cognitive decline. Therefore, we considered the effects of a regular exercise routine as a whole, which includes their interactive effects with other lifestyle-associated factors (i.e., indirect effects), and focused on the comprehensive effect of regular exercise routines, which is an interest in clinical practice.

However, some lifestyle-associated factors or even other factors could influence MEG spectral parameters independently from regular physical activity. We cannot exclude this possibility and cannot address these factors because this was a retrospective observational study: we do not have enough information about them. We described this as the third limitation (pages 27-28, lines 450-458).

Comment 1-3:

In my experience, the relationship between MEG results and cognitive function is considered to be difficult to understand.

Response:

Thank you for your comment. To address these doubts, we have further elaborated on the use of MEG as an assessment of cognitive decline (pages 5-6, lines 91-94 and page 14, lines 237-239).

We respect your view and understand that the majority of physicians have the impression that the use of MEG (and EEG, electrophysiological examinations in general) is difficult, except for patients with epilepsy. However, the comprehensibility of MEG results depends on the environment and strategies of clinical practice teams. We have been using MEG in the clinical practice for assessing patients with cognitive decline for 5 years. This has been enabled by the development of an analysis pipeline that reduces the analysis time/effort for technicians, by providing the quantification of the MEG measurements (e.g., MF, IAF, SSE, and others), and by summarising the results using a comprehensible format for all hospital staff member. We have been using this approach because we often receive feedback from hospital staff (i.e., physicians, clinical psychologists, clinical laboratory technicians, and therapists) that the results are clinically useful. Changes in MEG spectral parameters (i.e., MF, IAF, and SSE) are well matched with our clinical impressions [1]. We always show MEG data in two dimensional scatter plots along with patient data [2] or show their trends using line charts [1]. These are powerful tools to encourage patients and their families to change their lifestyle. Details of our clinical experiences have been published as a case series [1].

Comment 1-4:

Data availability is not clearly written. DOI is not effective.

Response:

We apologise the inconvenience. Unfortunately, we cannot reproduce the inaccessibility of the dataset using the doi: 10.17632/xgzgpk9475.1 (https://doi.org/10.17632/xgzgpk9475.1). Two of authors resolved the doi and verified that the link is working using independent environments. Could you please retry and let us know if it works?

Response to Reviewer 2 Comments

Comment 2-1

Can the authors provide references to explanations of values MF, IAF, SSE (l. 84).

Response:

Thank you for your question. According to this comment and comments by other reviewers, we have added a new subsection to describe the details of the spectral parameters (‘MEG analysis’ section; pages 12-14, lines 201-239). The pipelines for computing the MF, IAF, and SSE values are also extensively explained with references in this subsection.

Comment 2-2

Abbreviations in main text differ from those in the abstract (34-38) For example, in the abstract you note MMSE-J, but in the text just MMSE (53).

Response:

Thank you for your detailed reading and the note. Following the comment, we have carefully checked the wording throughout the manuscript. We intentionally used these two abbreviations with different meanings. We use ‘MMSE’ to refer to the Mini-Metal State Examination (in general) including original and translated versions, whereas we use ‘MMSE-J’ to specifically refer to its Japanese version which we used in the study. The validity and equivalence to the original version has already been evaluated [3]. Additionally, the copyright holders of the different versions are different; thus, we intestinally distinguished them.

Comment 2-3:

7, 111 – give corresponding statistics for men, especially statistics of age.

Response:

Thank you for the comment. The statistics (mean age and SD), which were described in the original manuscript, were not for women alone but included all patients. We apologies for the confusion. The number of men had been omitted in the original the manuscript, because we thought it would be inherently clear (total participants minus the number of women). Following the comment, we have added the number of the men (N = 139) (page 8, lines 138-140), but have not provided the statistics for each sex because this study did not focus on sex differences in any analyses.

Comment 2-4:

134-140. You here mention about special Japanese variant of corresponding scores. I think, you should provide concise description of differences them from international. In this context you should change abbreviations in the abstract.

Response:

Thank you for the comment. Following the comment, we have added relevant citations in the Methods section (page 11, lines 184-185). The MMSE-J and ADAS-J cog were officially translated to Japanese for the Alzheimer’s Disease Neuroimaging Initiative, which is an international collaboration project, and are widely used for both research and clinical medicine. The FAB-J has been validated as a translation of the original FAB and is widely used in Japan [4–6]. Therefore, we assume that there are no essential differences between the original and translated versions for all tests.

We apologise for the confusion caused by the description in the original abstract. We have modified it and explicitly described that we used their Japanese versions (page 2, lines 32-34).

Comment 2-5:

153 – abbreviation “meg” must be in uppercase.

Response:

Thank you for noting the typo. We have corrected it (page 15, line 255).

Comment 2-6:

In Methods you should describe hardware used for MEG abduction, protocol of this procedure, bands and channels you used for future analysis. The paper has complete lack of this technical information.

Response:

Thank you for the suggestion. Following the comment and comments by other reviewers, we have newly added subsections to describe the MEG data acquisition and analysis protocols (‘MEG data acquisition’ and ‘MEG analysis’ sections; pages 11-14, lines 191-239). We apologise for omitting these details in the original manuscript. Considering the topic of the present study, we assumed that few readers of this article would be interested in the details of the MEG techniques. However, the comments made us aware that these details are necessary for replications and required for scientific rigour. We hope that the necessary details are properly provided in these subsections; however, please kindly advise us again if you find that anything is still missing.

Comment 2-7:

I suppose a good idea would to provide short mathematical description of all parameters you analyze (MF, IAF, SSE).

Response:

Thank you for the comment. As replied to the previous comment (Comment #2-6), we have newly added a subsection to describe the MEG analysis protocols (‘MEG analysis’ section; pages 12-14, lines 201-239), where the explanations (including mathematical descriptions) of the MEG spectral parameters (i.e., MF, IAF, and SSE) have been also introduced.

Comment 2-8:

160-165. Provide mathematical formula for SOC value.

Response:

Thank you for the comment. We have added more detailed explanations regarding the SOC in the newly added subsection (‘Statistical analysis’ section; pages 15-16, lines 261-289). It is simply computed by dividing the group intercept (β0i) by the global slope (β1) of the linear regression model equivalent to the ANCOVA. We hope that the descriptions are sufficient to comprehend the SOCs, but we welcome further suggestions.

Comment 2-9:

173 – abbreviation “meg” must be in uppercase.

Response:

Thank you again for pointing this out. We have corrected it (page 18, lines 309-310).

Comment 2-10:

171 – 174. You write about significant differences in values. I think, you may provide here statistical results (p=…, etc.) mentioning the statistical method of evaluation.

Response:

Thank you for your suggestion. Following the comment, we have added the p-values in the main text (pages 17-18, lines 306-311). We intentionally omitted them from the original manuscript, because we listed them in Table 1. However, they were not properly displayed in the first version of the manuscript, due to a formatting error. We feel that the statistical values would provide support to the main text; thus, we have added them.

Comment 2-11:

181. Please provide more informative header of the table. What is P(unk)? In the text above (l. 173-179) you mention Spearman’s rank correlation but the table doesn’t provide any information about it. By the way the Table 1 provides information about groups of patients. I think you should include one more table with result of Spearman’s rank correlation analysis.

Response:

Thank you for the comments and suggestions. Following the comment and comments by other reviewers, we have modified the Results section and Table 1, and newly added S1 and S2 Tables (‘Group-level differences and correlations’ subsection; pages 17-18, lines 303-322). As correctly pointed out, Table 1 in the original manuscript was not informative because insufficient information was available to comprehend the contents (also, it seems that a few columns were missing at the right side of the table, possibly due to incorrect formatting). We apologise for this confusion. We have modified Table 1, so that it includes the results regarding the descriptive statistics of the parameters for each group (i.e., ‘Active’ and ‘Nonactive’) and their group-level comparisons using t-tests. Furthermore, we have added new tables (S1 and S2 Tables) to summarise the results of the correlation analyses. The results of the correlation analyses were described in the main text in the original manuscript; thus, they were omitted in the tables. However, it would be more comprehensible to summarise them in the table format; so we have added these tables in the revised manuscript. Please note that the statistical approach has been changed from nonparametric (Wilcoxon’s test and Spearman’s correlation) to parametric tests (t-test and Pearson’s correlations), given the large number of samples (N = 327) (pages 14-15, lines 241-260) and following a suggestion from another reviewer.

Comment 2-12:

Again, 188 - abbreviation “meg” must be in uppercase. Please look through the text and correct this error.

Response:

Thank you again. We apologies for the many typos. We have removed this line from the revised manuscript but have carefully proofread the whole manuscript to correct any remaining typos.

Comment 2-13:

188-191. Excuse me but I see no strong correlation between age and scores. There mostly are horizontal linear regression lines. I am strongly against “ritual” phrases in statistical reports (l. 191). If you mention ANOVA you should describe statistical model, you used, like one-way, two-way, repeated measurement and so on. Statistical results should have F and p values and information about post-hoc estimation of pair-wise comparison.

Response:

Thank you for pointing this out. Following the comments, we have modified the Results section (page 18, lines 312-314 and page 20, lines 332-334) and Table 2, and newly added S1 and S2 Tables. Regarding the correlation, we found no associations (i.e., correlational or regression relationships) between age and MEG spectral parameters. We did n

---

## [Decision Letter · Decision Letter 1]

PONE-D-24-41991R1Regular physical activity affects brain activities in old individuals: an observational studyPLOS ONE

Dear Dr. Shigihara,

Thank you for submitting your manuscript to PLOS ONE. After careful consideration, we feel that it has merit but does not fully meet PLOS ONE’s publication criteria as it currently stands. Therefore, we invite you to submit a revised version of the manuscript that addresses the points raised during the review process.

Reviewers generally appreciated the authors' efforts to clarify the methods and improve the coherence of the manuscript. However, there is still substantial ambiguity regarding a crucial part of the methods—specifically, the inclusion criteria and the population studied. One reviewer highlighted that the tests used may not be appropriate for healthy aging subjects. It remains unclear whether a mix of healthy subjects and patients with dementia was studied, and if so, what the rationale was for this selection. Please clearly specify the population studied, including diagnoses, and clarify the reasoning behind the choice of tests. Note that the manuscript may not be accepted if the population selection is not coherent with the study’s aim and relevant methods

We look forward to receiving your revised manuscript.

Kind regards,

Federico Giove, PhD

Academic Editor

PLOS ONE

Reviewers' comments:

Reviewer's Responses to Questions

**Comments to the Author**

1. If the authors have adequately addressed your comments raised in a previous round of review and you feel that this manuscript is now acceptable for publication, you may indicate that here to bypass the “Comments to the Author” section, enter your conflict of interest statement in the “Confidential to Editor” section, and submit your "Accept" recommendation.

Reviewer #1: (No Response)

Reviewer #2: (No Response)

Reviewer #3: All comments have been addressed

Reviewer #5: All comments have been addressed

2. Is the manuscript technically sound, and do the data support the conclusions?

Reviewer #1: No

Reviewer #2: Yes

Reviewer #3: Yes

Reviewer #5: Yes

3. Has the statistical analysis been performed appropriately and rigorously? 

Reviewer #1: No

Reviewer #2: Yes

Reviewer #3: Yes

Reviewer #5: Yes

4. Have the authors made all data underlying the findings in their manuscript fully available?

Reviewer #1: Yes

Reviewer #2: Yes

Reviewer #3: Yes

Reviewer #5: Yes

5. Is the manuscript presented in an intelligible fashion and written in standard English?

Reviewer #1: Yes

Reviewer #2: Yes

Reviewer #3: Yes

Reviewer #5: Yes

6. Review Comments to the Author

Reviewer #1: The manuscript described the daily activities affecting the mental function in aged.

There are several problems, and among them I wold like to point out the psychological examinations; adopted tests are for the medical diagnostic test for dementia. It is necessary to measure the normal persons maintaining the brain functions, such as Wechsclar memory scale for memory, Trail making tests for attention, Barthel index for ADL and so on. The psychological examination for people of good normal life could not be evaluated by medical psychological examinations such as muse, or others.

In order to evaluate the brain integrity, it is difficult to do by MEG. In a reverse sense, it is expected that the aged brain have a MEG or EEG function on a particular pattern or not.

In my impression, the design of the study is not good for the intended purpose.

Reviewer #2: All is well. I have only two small comments.

l. 329 “ANCOVA and SOC” The title of the section does not give any information about the topic.

I think a better idea is from your text below "Effect of groups on MEG spectral parameters".

l. 357 Table 2. Results of Analysis of Covariance. I already pointed out this inaccuracy in the previous review (see comment 2-15). It is better to give the table title from your text “Assessing the effects of regular physical activity on MEG spectral parameters”. In your reply to me, you wrote that you had corrected this inaccuracy, but apparently you forgot to do so in the text.

Reviewer #3: I thank the authors for addressing my comments of the previous round. The manuscript has improved as a result. I don’t have any further comments.

Reviewer #5: Minor points

1) Line 56. Typo. Its process -> This process

2) Line 62. Typo. Influence on pathological conditions -> influence pathological conditions

3) Line 85-86. Which are sensitive to the cognitive decline -> which are sensitive to cognitive decline

4) Figure 1. Bottom left corner. Week association -> Weak association

5) Line 100. and considered drivers of cognitive impairments -> and are considered drivers of cognitive impairments

6) Line 373. These factors interfere each other -> these factors Interfere with each other

7) Lines 375 and 378 as the direct/indirect pathways -> as direct/indirect pathways

8) Line 381. From a clinical perspective it is not essential whether… -> From a clinical perspective, it is not clear whether…

9) Line 385. This study showed that MEG spectral parameters were enhanced by regular physical activity ----- Enhanced might be an inappropriate word choice here. Consider using less causal language.

10) Line 388. Contain -> Reflect

11) Line 425. Theoretically, an increase in high-frequency oscillatory activity corresponds to high MF, IAF, and SSE values. ----- This is highly speculative, as spectral parameters are jointly modulated by a myriad of neurophysiological and neuromodulatory phenomena, rather than oscillatory power within a single frequency band. I would suggest the authors to either remove this claim or substantiate it with empirical evidence.

Final comments

I would like to thank the authors for taking the time to implement the suggested corrections and submit a much-improved version of their original work. I enjoyed reading this new version and found their line of argumentation much clearer. The description of data analysis and statistical procedures is much clearer and logically responds to their research question. I also appreciate the inclusion of MEG data acquisition and preprocessing procedures. I am still somewhat confused as to why the authors chose to use SOC to quantify the effect of physical activity on predicting MEG spectral parameters, rather than more conventional methods such as mediation analyses. However, I do appreciate the mathematical and statistical explanation provided by the authors, which allowed me to understand their approach better than in the original submission. The only thing I am missing from this new version is a more careful explanation of what the different spectral parameters included in this study measure, neurophysiologically speaking (i.e., changes in excitation/inhibition ratios, changes in neuromodulator release, power ratios between different frequency bands, etc.). This would help the reader make a smoother connection between physical activity, brain physiology, and cognition.

7. PLOS authors have the option to publish the peer review history of their article (what does this mean? ). If published, this will include your full peer review and any attached files.

**Do you want your identity to be public for this peer review?** For information about this choice, including consent withdrawal, please see our Privacy Policy .

Reviewer #1: **Yes: ** OK

Reviewer #2: **Yes: ** Andriy Gorkovenko

Reviewer #3: No

Reviewer #5: No

---

## [Author Response · Author response to Decision Letter 2]

24 Mar 2025

Response to reviewers:

Regular physical activity affects brain activities in old individuals:

an observational study

Short title: Physical activity and brain activity

Keisuke Fukasawa1, ¶, Hideyuki Hoshi2,3, ¶, Yoko Hirata4, Momoko Kobayashi3, Keita Shibamiya3, Sayuri Ichikawa1, Yoshihito Shigihara2,3*

1Clinical Laboratory, Kumagaya General Hospital, Kumagaya City, Saitama, Japan,

2Precision Medicine Centre, Hokuto Hospital, 7-5 Kisen, Inada-cho, Obihiro-shi, Hokkaido, Japan

3Precision Medicine Centre, Kumagaya General Hospital, Kumagaya City, Saitama, Japan

4Department of Neurosurgery, Kumagaya General Hospital, Kumagaya City, Saitama, Japan

* Corresponding author Yoshihito Shigihara

E-mail: y-shigihara@hokuto7.or.jp (YS)

¶These authors contributed equally to this work.

To the editor and reviewers:

Thank you for another opportunity to submit a revised draft of our manuscript titled ‘Regular physical activity affects brain activities in old individuals: an observational study’.

We appreciate the time and effort you have dedicated to providing valuable feedback on our manuscript. We are grateful for your insightful comments that helped us improve the quality of our study. We have incorporated changes to reflect all your suggestions.

Please find our point-by-point responses to all comments and concerns below.

Role of Funder statement

This study was partially supported by RICOH Co., Ltd. The funders had no role in study design, data collection and analysis, decision to publish, or preparation of the manuscript.

Response to Editor

Reviewers generally appreciated the authors' efforts to clarify the methods and improve the coherence of the manuscript. However, there is still substantial ambiguity regarding a crucial part of the methods—specifically, the inclusion criteria and the population studied. One reviewer highlighted that the tests used may not be appropriate for healthy aging subjects. It remains unclear whether a mix of healthy subjects and patients with dementia was studied, and if so, what the rationale was for this selection. Please clearly specify the population studied, including diagnoses, and clarify the reasoning behind the choice of tests. Note that the manuscript may not be accepted if the population selection is not coherent with the study’s aim and relevant methods

Response:

Thank you for your efforts in editing our manuscript with the reviewers. We understand that the editor raised two concerns: (1) inclusion criteria and (2) target population of the study.

Regarding the inclusion criteria, we have explicitly described the inclusion and exclusion criteria (page 8, lines 139 — 144). This study was conducted to improve clinical MEG examination and provide information for patients’ treatments (pages 6 — 7, lines 110 — 112; page 29, lines 492 — 493). Thus, we designed the study to have the clinical population as real as possible. Therefore, we included almost all patients who visited our outpatient section of dementia care to seek treatment, regardless of their diagnosis (page 9, lines 147 — 149). Lifestyle modification is effective for almost all individuals (See WHO’s risk reduction of cognitive decline and dementia [1]), and there is little benefit in distinguishing them by their diagnoses. We did not mention their clinical diagnoses in the previous version of the manuscript because they are not directly relevant to the scope of the present study. However, the information regarding the diagnosis would allow readers to grasp the quality of the clinical population of the patients; we have added them in the revised manuscript (pages 8 — 9, lines 144-147).

Regarding the study population, as mentioned above, we included as many patients as possible, regardless of their diagnoses (page 9, lines 147 — 149). Reviewer 1 pointed out that ‘the tests used may not be appropriate for healthy aging subjects.’ However, we consider this point to be inapplicable to the present study for four reasons. First, neither ‘mental function’ nor ‘diagnose’ is within the scope of the present study. We are afraid that Reviewer 1 might misunderstand the subject and purpose of the present study. The subject of the present study is ‘brain activity’ and not ‘mental function.’ We have explicitly described this in the title of the manuscript and in lines 21–22 and 110 – 112. The motivation of the present study is to improve clinical MEG to provide better clinical practice for patients who are afraid of dementia. To achieve this goal, we investigated brain activity measured using MEG to evaluate the influence of regular physical activity, which is a modifiable factor for preventing dementia. Although we used neuropsychological assessments (MMSE, FAB, and ADAS-cog) in the present study, they were merely used to adjust the influence of cognitive impairment (i.e., confounding factors) on brain activity; they were not the core of the study. Second, the selection of neuropsychological assessments (MMSE, FAB, and ADAS-cog) as quantifications of the confounding factors is reasonable for the following reasons. (2-1) A previous study has shown that MMSE and FAB scores influence the MF, IAF, and SSE values [2] (page 6, lines 95 — 96). It is unclear whether the Wechsclar memory scale, Trail making tests, and Barthel index influence MEG spectral parameters, although they are likely to do so. (2-2) We intend to use the present results in clinical practice. We should select neurophysiological assessments that are frequently used in the outpatient section for dementia care. We selected the three assessments routinely used in our hospital (Page 11, lines 188 — 191). They are also widely used to screen patients to distinguish individuals who are at risk from healthy individuals. This indicates that the assessments are applicable to both healthy individuals and patients with cognitive impairment. Third, the study protocol used in the present study was also used in our previous studies, which included both healthy individuals and patients with cognitive impairment and dementia [2,3], which supported the validity of the study. Fourth, healthy conditions and cognitive impairment are a continuum [4]. There was no gap between healthy individuals and patients with cognitive impairment. It is unreasonable to claim that the three assessments cannot be used for healthy individuals. Although we appreciate Reviewer 1’s detailed reading and careful evaluation, we consider that the comment would be beside the point.

We believe that our study setting is the best (available) approach, and the present study contributes to the happiness of future patients.

Response to Reviewer 1 Comments

The manuscript described the daily activities affecting the mental function in aged.

There are several problems, and among them I wold like to point out the psychological examinations; adopted tests are for the medical diagnostic test for dementia. It is necessary to measure the normal persons maintaining the brain functions, such as Wechsclar memory scale for memory, Trail making tests for attention, Barthel index for ADL and so on. The psychological examination for people of good normal life could not be evaluated by medical psychological examinations such as muse, or others.

In order to evaluate the brain integrity, it is difficult to do by MEG. In a reverse sense, it is expected that the aged brain have a MEG or EEG function on a particular pattern or not.

In my impression, the design of the study is not good for the intended purpose.

Response:

We appreciate the reviewer’s detailed reading and careful evaluation. We are afraid that the reviewer might misunderstand the subject and purpose of the present study. Although they commented that ‘The manuscript described the daily activities affecting the mental function in aged’, mental function is not our main interest. We are interested in the relationship between regular physical activity and brain activity, as described in the title of this manuscript.

Furthermore, we would like to remind that the present study did not focus on the distinction between diagnostic groups (e.g., healthy controls vs. patients with dementia). The motivation for the present study was to improve clinical MEG examination to provide better practice for patients who are afraid of dementia. The retrospective study included a dataset of all eligible patients who visited the outpatient section of dementia to replicate the real clinical population, while their diagnostic and pathological backgrounds were not considered as inclusion/exclusion criteria. Lifestyle modification is effective for everyone, regardless of their diagnosis or clinical state, not only for primary prevention of cognitive impairment but also for secondary and tertiary prevention (See WHO’s risk reduction of cognitive decline and dementia [1]). Thus, it is essential to include all individuals who visited our hospital to seek medical care, regardless of their cognitive state. Therefore, they did not need to be screened or categorised strictly using detailed examinations for the purpose of the present study. To be explicit about the patients’ clinical backgrounds and allow readers to grasp the quality of the clinical population of the patients, we have added these information in the revised manuscript (pages 8 — 9, lines 144 — 147).

Essentially, healthy conditions and cognitive impairment are a continuum [4]. There is no gap between healthy individuals and patients with cognitive impairment. All individuals, especially older individuals, are at risk of dementia; there are no perfectly healthy individuals. Therefore, the claim that the three assessments (MMSE, FAB, and ADAS) cannot be used for healthy individuals in clinical practice is unreasonable. This is supported by the figures in our previous studies [2,3][Figure 2 in Hoshi et al. (2022) [2] and Figure 4 in Hoshi et al. (2023) [3]]. These studies included healthy individuals and patients with cognitive impairment and dementia. They used a strategy similar to the present one and showed that MMSE and FAB can be applied to both healthy individuals and patients with cognitive impairment and dementia.

We used neuropsychological assessments, such as the MMSE, to control for the influence of cognitive level on brain activity. Therefore, ‘mental function’ was a confounding factor in this study. If the main subject of the present study was associated with ‘mental function’, we would agree with the reviewer that the function should be assessed more extensively. However, it is not. We considered the choice of using the three assessments as quantifications of the confounding factors reasonable for the following reasons. First, a previous study has shown that MMSE and FAB scores influence MF, IAF, and SSE values [2] (page 6, lines 95 — 96). It is unclear whether the Wechsclar memory scale, Trail making tests, and Barthel index influence MEG spectral parameters, although they are likely to do so. Second, we routinely use the three neuropsychological assessments in our outpatient department for dementia treatment. We also use other neuropsychological assessments, but not routinely. The motivation of the present study is to improve cognitive treatment for cognitive impairment and dementia; thus, the frequently used assessments were naturally the focus of the study.

As a summary, if the main subject of the present study was associated with ‘mental function’, we would agree with the reviewer that the function should be assessed more extensively. However, it is not. Therefore, we believe that the present study design is reasonable and practical for addressing the present subject.

Response to Reviewer 2 Comments

l. 329 “ANCOVA and SOC” The title of the section does not give any information about the topic.

I think a better idea is from your text below "Effect of groups on MEG spectral parameters".

Response:

Thank you very much for your constructive suggestions. Following the comment, we have changed the title of the subsection as suggested (page 22, line 359).

Response:

l. 357 Table 2. Results of Analysis of Covariance. I already pointed out this inaccuracy in the previous review (see comment 2-15). It is better to give the table title from your text “Assessing the effects of regular physical activity on MEG spectral parameters”. In your reply to me, you wrote that you had corrected this inaccuracy, but apparently you forgot to do so in the text.

Thank you for your suggestion. Following the comment, we have modified the title of Table 2 as suggested (page 23, line 386). We apologise that our modification in the previous round of review was not satisfactory to the reviewer.

Response to Reviewer 5 Comments

Minor points

1) Line 56. Typo. Its process -> This process

2) Line 62. Typo. Influence on pathological conditions -> influence pathological conditions

3) Line 85-86. Which are sensitive to the cognitive decline -> which are sensitive to cognitive decline

4) Figure 1. Bottom left corner. Week association -> Weak association

5) Line 100. and considered drivers of cognitive impairments -> and are considered drivers of cognitive impairments

6) Line 373. These factors interfere each other -> these factors Interfere with each other

7) Lines 375 and 378 as the direct/indirect pathways -> as direct/indirect pathways

8) Line 381. From a clinical perspective it is not essential whether… -> From a clinical perspective, it is not clear whether…

9) Line 385. This study showed that MEG spectral parameters were enhanced by regular physical activity ----- Enhanced might be an inappropriate word choice here. Consider using less causal language.

10) Line 388. Contain -> Reflect

Response:

We appreciate the reviewer for the detailed reading and thank you very much for finding typos and making suggestions. Following your comments, we have modified/corrected them. Regarding comment 9), we used ‘associated’ instead of ‘enhanced’ to describe the indirect and non-causal relationships between the MEG spectral parameters and regular physical activity (page 26, line 418).

11) Line 425. Theoretically, an increase in high-frequency oscillatory activity corresponds to high MF, IAF, and SSE values. ----- This is highly speculative, as spectral parameters are jointly modulated by a myriad of neurophysiological and neuromodulatory phenomena, rather than oscillatory power within a single frequency band. I would suggest the authors to either remove this claim or substantiate it with empirical evidence.

Response:

Thank you for your constructive comments. Following this comment, we removed the sentence (page 28, line 455).

Final comments

I would like to thank the authors for taking the time to implement the suggested corrections and submit a much-improved version of their original work. I enjoyed reading this new version and found their line of argumentation much clearer. The description of data analysis and statistical procedures is much clearer and logically responds to their research question. I also appreciate the inclusion of MEG data acquisition and preprocessing procedures. I am still somewhat confused as to why the authors chose to use SOC to quantify the effect of physical activity on predicting MEG spectral parameters, rather than more conventional methods such as mediation analyses. However, I do appreciate the mathematical and statistical explanation provided by the authors, which allowed me to understand their approach better than in the original submission.

Response:

Thank you for your kind message. Regarding the employment of SOC, we used the strategy to quantify the effect of regular physical activity using the language (i.e., unit) of neuropsychological assessment but not that of neurological parameters (i.e., MEG spectral parameters). For clinicians involved in the care of patients with cognitive decline, it would be difficult to comprehend the significance of the results, such as ‘the patients with regular physical exercise showed 2.0 point higher MF than those without’, because electrophysiological examinations are not in the mainstream of the clinical practice of cognitive decline. Therefore, we attempted to convert it using the SOC and described that ‘the patients with regular physical exercise showed higher MF tha

---

## [Editor Report · Decision Letter 2]

PONE-D-24-41991R2

Regular physical activity affects brain activities in old individuals: an observational study

PLOS ONE

Dear Dr. Shigihara,

Thank you for submitting your manuscript to PLOS ONE. After careful consideration, we have decided that your manuscript does not meet our criteria for publication and must therefore be rejected.

Specifically:

The study suffers from significant sampling bias . The studied sample is not representative of a specific population (e.g. healthy elderly subjects, early AD, etc), but simply reflects the people that seek consulting at the involved Medical Centre. This is not a meaningful population and can’t be easily generalized.

Moreover, only in the last revision authors listed the outcome of the diagnostic path, stating that among the subjects “32 were diagnosed with healthy ageing, 60 with mild cognitive impairment, and 220 with dementia”. The sample was treated as a whole, irrespectively of health condition, and the statistical analysis contrasted active vs non-active, with no covariate related to diagnosis. Instead, neuropsychological assessments were used as covariates, and they may be inappropriate across the involved sub-populations. Irrespectively of the covariates, this approach obviously prevented the study of interactions activity x health condition. It may well be that the reported  effect is dominated by one sub-group.

Finally, the manuscript, including the title, refers generally to “old individuals”, and states that findings can be used in patients with cognitive impairment. All of these are undue generalizations, if the sampled population does not match the claims.

I’m sorry this fundamental problem was missed in the first round of reviews and was catch only during the second.

The manuscript can be resubmitted, but only if the mentioned problems are addressed. As it stands, the methods are not appropriate

I am sorry that we cannot be more positive on this occasion, but hope that you appreciate the reasons for this decision.

Kind regards,

Federico Giove, PhD

Academic Editor

PLOS ONE

- - - - -

---

## [Author Response · Author response to Decision Letter 3]

12 Apr 2025

Response to reviewers (for Appeal):

Regular physical activity affects brain activities in old individuals:

an observational study

Short title: Physical activity and brain activity

ID: PONE-D-24-41991R2

Keisuke Fukasawa1, ¶, Hideyuki Hoshi2,3, ¶, Yoko Hirata4, Momoko Kobayashi3, Keita Shibamiya3, Sayuri Ichikawa1, Yoshihito Shigihara2,3*

1Clinical Laboratory, Kumagaya General Hospital, Kumagaya City, Saitama, Japan,

2Precision Medicine Centre, Hokuto Hospital, 7-5 Kisen, Inada-cho, Obihiro-shi, Hokkaido, Japan

3Precision Medicine Centre, Kumagaya General Hospital, Kumagaya City, Saitama, Japan

4Department of Neurosurgery, Kumagaya General Hospital, Kumagaya City, Saitama, Japan

* Corresponding author Yoshihito Shigihara

E-mail: y-shigihara@hokuto7.or.jp (YS)

¶These authors contributed equally to this work.

Comment 1.

The study suffers from significant sampling bias. The studied sample is not representative of a specific population (e.g. healthy elderly subjects, early AD, etc), but simply reflects the people that seek consulting at the involved Medical Centre. This is not a meaningful population and can’t be easily generalized.

Response 1:

While we appreciate the careful review of and detailed feedback regarding our manuscript, we feel that the word ‘bias’ was used incorrectly here. As we explicitly described in the manuscript (lines from 110–112) and also clarified in our responses to the reviewers, the goal of the present study was to develop a clinical tool to facilitate the assessment of lifestyle-associated factors in clinical environments. We strongly believe that, to achieve this purpose, the study cohort must comprise patients who seek lifestyle intervention (lines from 147–149). Therefore, we have used the presented dataset which included patients who visited a hospital for that purpose. While the editor stated that the effects should be examined in a group of patients with/without specific diagnostic (pathological) backgrounds, we believe that such a selection would actually bias our sample. In general, we would be happy to perform additional analyses of interest within subgroups of our cohort; however, considering the original goal of our study, we do not see how such additional analyses would fit within the scope of the current manuscript. We would appreciate if the editor could be clearer regarding the reason for recommending using datasets from patients subgroups to address the purpose of the study.

Comment 2.

Moreover, only in the last revision authors listed the outcome of the diagnostic path, stating that among the subjects “32 were diagnosed with healthy ageing, 60 with mild cognitive impairment, and 220 with dementia”. The sample was treated as a whole, irrespectively of health condition, and the statistical analysis contrasted active vs non-active, with no covariate related to diagnosis. Instead, neuropsychological assessments were used as covariates, and they may be inappropriate across the involved sub-populations. Irrespectively of the covariates, this approach obviously prevented the study of interactions activity x health condition. It may well be that the reported effect is dominated by one sub-group.

Response 2:

As we previously responded to the reviewers (lines from 47–48, 66–68, 115–117, and 149–151) in the second round, clinical diagnoses were not within the scope of the present study. The goal of the present study was to develop a clinical tool to assess lifestyle-associated factors in clinical environments. In real-life diagnostic processes in the clinical practice, diagnoses follow examinations; i.e., at the time of clinical examinations, including MEG, there are no established diagnoses. Clinicians make diagnoses based on medical interviews and clinical examinations. The results of the MEG examinations, including the quantification of the influence of regular physical activity on brain activity, are considered during the diagnosing process. Therefore, for evaluating the influence of regular physical activity on MEG spectral parameters, diagnosis information is not available. Consequently, we believe that the editor’s suggested model including diagnostic information would be unsuitable in the clinical practice. In contrast, neuropsychological assessment scores are usually available at the time of MEG examinations, because these assessments may happen on the same day of the MEG scan, which always precedes the diagnosing day. Keeping the purpose of the present study and the actual clinical environment in mind, we hope that the editor will agree with the suitability of our presented approach.

Comment 3

Finally, the manuscript, including the title, refers generally to “old individuals”, and states that findings can be used in patients with cognitive impairment. All of these are undue generalizations, if the sampled population does not match the claims.

Response 3:

We strongly disagree with this comment. To generalize the results to “old individuals”, the studied population must include old individuals from all backgrounds without any further prerequisites. The goal of the present study was to develop a clinical tool to assess lifestyle-associated factors in clinical environments (lines 47–48, 66–68, 115–117, and 149–151 in the response letter of the second round). The study cohort comprised older individuals who visited a hospital to seek medical care, which genuinely represents “old individuals”. We would appreciate if the editor could be more explicit as to why they believe that the study population did not match “old individuals” and explain which populations would satisfies their expectation.

---

## [Decision Letter · Decision Letter 3]

Regular physical activity affects brain activities in old individuals: an observational study

PONE-D-24-41991R3

Dear Dr. Yoshihito Shigihara,

We’re pleased to inform you that your manuscript has been judged scientifically suitable for publication and will be formally accepted for publication once it meets all outstanding technical requirements.

Kind regards,

Hiroki Annaka

Academic Editor

PLOS ONE

Additional Editor Comments (optional):

I have re-evaluated your paper as a new Academic Editor. I have judged your paper as accepted through peer review.

Reviewers' comments:

Reviewer's Responses to Questions

**Comments to the Author**

1. If the authors have adequately addressed your comments raised in a previous round of review and you feel that this manuscript is now acceptable for publication, you may indicate that here to bypass the “Comments to the Author” section, enter your conflict of interest statement in the “Confidential to Editor” section, and submit your "Accept" recommendation.

Reviewer #1: All comments have been addressed

Reviewer #2: All comments have been addressed

2. Is the manuscript technically sound, and do the data support the conclusions?

Reviewer #1: Yes

Reviewer #2: Yes

3. Has the statistical analysis been performed appropriately and rigorously? 

Reviewer #1: Yes

Reviewer #2: Yes

4. Have the authors made all data underlying the findings in their manuscript fully available?

Reviewer #1: Yes

Reviewer #2: Yes

5. Is the manuscript presented in an intelligible fashion and written in standard English?

Reviewer #1: Yes

Reviewer #2: Yes

6. Review Comments to the Author

Reviewer #1: The comments by reviewers were well revised and methodology (MEG, statistical analyses), as well as good described results and discussion improved, those satisfied Plos One criteria. I hope prospective study in the future will produce more fruitful results, meaning more strict scientific results anticipated.

Reviewer #2: You corrected my remarks and I have no complaints against you. I recommend that the editor publish your article.

7. PLOS authors have the option to publish the peer review history of their article (what does this mean? ). If published, this will include your full peer review and any attached files.

**Do you want your identity to be public for this peer review?** For information about this choice, including consent withdrawal, please see our Privacy Policy .

Reviewer #1: **Yes: ** H. Fukuyama

Reviewer #2: **Yes: ** Andriy V. Gorkovenko

---

## [Editor Report · Acceptance letter]

PONE-D-24-41991R3

PLOS ONE

Dear Dr. Shigihara,

I'm pleased to inform you that your manuscript has been deemed suitable for publication in PLOS ONE. Congratulations! Your manuscript is now being handed over to our production team.

Kind regards,

on behalf of

Dr. Hiroki Annaka

Academic Editor

PLOS ONE